# Task-specific modulation of corticospinal neuron activity during motor learning in mice

Najet Serradj[1,6], Francesca Marino[2,6], Yunuen Moreno-López[1], Amanda Bernstein[1], Sydney Agger[3], Marwa Soliman[1], Andrew Sloan [4] & Edmund Hollis [1,5] ✉

Motor skill learning relies on the plasticity of the primary motor cortex as task acquisition drives cortical motor network remodeling. Large-scale cortical remodeling of evoked motor outputs occurs during the learning of corticospinal-dependent prehension behavior, but not simple, non-dexterous tasks. Here we determine the response of corticospinal neurons to two distinct motor training paradigms and assess the role of corticospinal neurons in the execution of a task requiring precise modulation of forelimb movement and one that does not. In vivo calcium imaging in mice revealed temporal coding of corticospinal activity coincident with the development of precise prehension movements, but not more simplistic movement patterns. Transection of the corticospinal tract and optogenetic regulation of corticospinal activity show the necessity for patterned corticospinal network activity in the execution of precise movements but not simplistic ones. Our findings reveal a critical role for corticospinal network modulation in the learning and execution of precise motor movements.

The corticospinal tract is the principal mediator of dexterous movements that require precise execution and adaptive modulation; however, the effects of dexterous prehension training on corticospinal neuron activity remain unknown. The learning of dexterous movements depends on both the integrity of the corticospinal tract as well as the plasticity of cortical motor networks. Attenuation of cortical plasticity impairs the learning, but not the execution of previously learned, complex motor behaviors[1]. The plasticity of these cortical motor networks has been studied, with stimulation of individual cortical points able to elicit distinct movements[2]. Electrical microstimulation in primate and rodent cortex has been used to demonstrate the labile nature of individual cortical sites[3] and how both learning, and injury can reshape motor representations, or maps[4–6]. As rodents learn a forelimb prehension task requiring precise motor control, motor representations of the musculature involved in the trained movement increase, extending into neighboring map areas[1,7].

Additionally, precision prehension training drives enhanced correlation between firing rate in the primary motor cortex and muscle recruitment patterns and more reliable muscle recruitment with improved performance[8]. This large-scale physiological reorganization does not occur in animals trained on a non-dexterous lever-press task[7].

The distinction between precise and imprecise behavior is a critical one, as both learning and execution of precise, dexterous movements are dependent upon corticospinal output. Damage to the motor cortex or transection of the corticospinal tract in both primates and rodents impairs the execution of previously trained, dexterous forelimb movements[9–13]. In non-human primates, lesioning the corticospinal tract at the medullary pyramids (pyramidotomy) results in a permanent loss of precision grip and dexterous hand control, with only transient impact on less dexterous grasp ability[11]. Similarly, pyramidotomy in rodents permanently impairs dexterous forelimb control, resulting in impaired reaching and grasping behavior[9]. In contrast,

[1]Burke Neurological Institute, White Plains, NY, USA. [2]University of California, San Francisco, CA, USA. [3]HAVAS Production Studios, New York, NY, USA. [4]Vulintus, Inc., Lafayette, CO, USA. [5]Feil Family Brain and Mind Research Institute, Weill Cornell Medicine, New York, NY, USA. [6]These authors contributed equally: Najet Serradj, Francesca Marino. ✉e-mail: edh3001@med.cornell.edu

once trained, the engrained movements on a simple lever press task can be faithfully executed by rodents in the absence of the entire motor cortex[14]. Motor cortex plays a key instructional role during the early phases of lever press learning; however, the development of expertise leads to disengagement of motor cortex from this imprecise movement[15]. Silencing of motor cortex in rodents during early lever press training impairs performance, while eliciting little effect after long-term training[16], likely owing to a larger role for subcortical motor circuits in trained animals. In contrast, following training on the precise single pellet reach behavior, there appears to be a dissociation of gross motor movements from fine control, with a greater reliance on motor cortex than basal ganglia for mediating precision control of forelimb grasp[17]. It is likely that complex, dexterous behaviors require an active pattern of sensory feedback for fine-tuning forelimb movements and error correction[18].

There are structural changes that occur in corticospinal neurons during dexterous motor learning that support a role for the direct involvement of these layer 5 pyramidal neurons in task acquisition. In vivo imaging of layer 5 neurons has revealed a rapid induction of dendritic spine formation while learning dexterous tasks. In these neurons, successively formed spines cluster along the dendrite, which could act to amplify the post-synaptic response to related task-specific inputs[19]. Using selective labeling of corticospinal neurons projecting to spinal levels responsible for distal forelimb control, dexterous motor learning has been shown to drive selective learning-dependent increases in spine density and branching in comparison to neighboring neurons that control proximal forelimb movements[20]. These structural changes occur in parallel with a period of increased spine dynamics and clustering of newly formed task-related spines in more superficial excitatory layer 2/3 neurons during learning[21] suggestive of a strengthening of behaviorally relevant motor circuitry.

To determine the role of corticospinal neurons in the cortical reorganization that occurs during dexterous motor learning, we combined in vivo calcium imaging with training on both precise (precision) and imprecise (adaptive) versions of a forelimb isometric pull task. Both tasks utilize similar forelimb movements and are therefore likely to engage the same population of corticospinal neurons; however, only the precision movement was found to depend on the integrity of the corticospinal tract. We tracked forepaw corticospinal neuron activity in the primary motor cortex over the course of task learning and found critical differences in the response to precision and adaptive learning. The development of expertise occurred rapidly during adaptive training and showed no significant correlation between corticospinal activity and movement kinetics; whereas the development of precision expertise drove a refinement of movement kinetics and corresponding increase in task-associated neuron activity. Animals that failed to develop expertise in the precision group exhibited a more limited repertoire of dynamic movements and a corresponding absence of network modulation. We demonstrate that these learning-mediated changes depend on the spinal connections of the corticospinal tract as transection at the level of the medullary pyramids disrupts corticospinal networks and learned movements. Finally, altering the pattern of corticospinal network activity impairs the execution of precise movements but not the imprecise movements on the adaptive task. These findings illustrate the key role of corticospinal neurons in the learning and execution of dexterous forelimb movements.

## Results

### Dynamic movement adaptation occurs during precision learning

To determine the role of corticospinal neurons in the cortical reorganization that occurs during precise, but not imprecise, motor learning, we modified the MotoTrak isometric pull task (Vulintus, Inc.)[22] to require precise modulation of forelimb movements. We employed both imprecise (adaptive) and precise (precision) versions of the task in combination with two-photon microscopy in head-fixed mice (Fig. 1a–c).

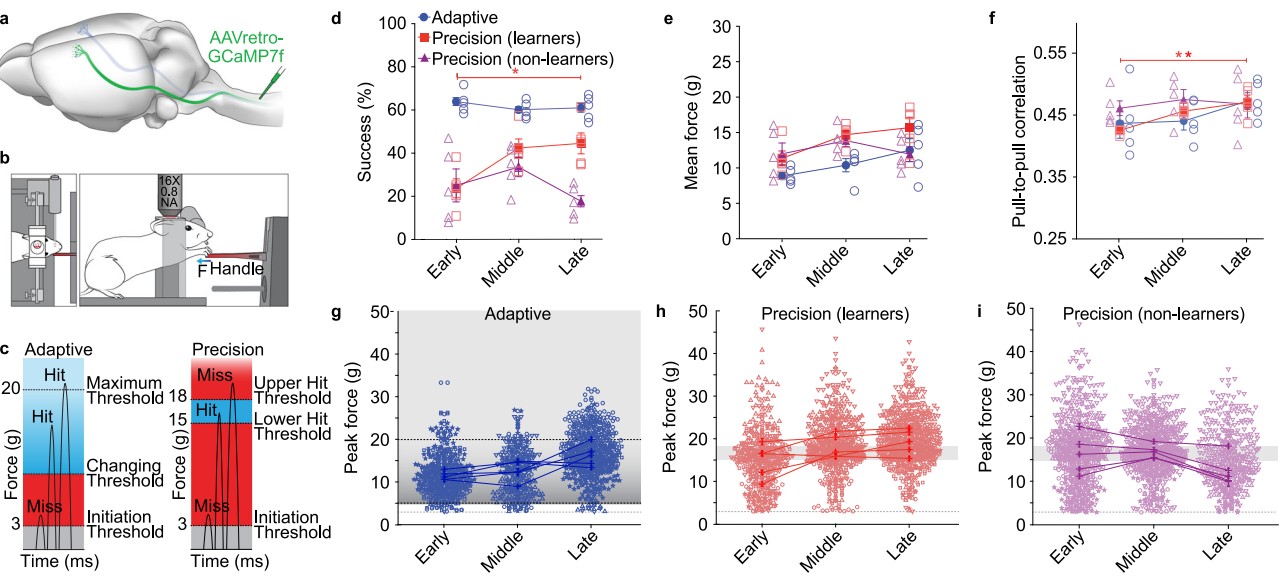

**Fig. 1 | Dynamic movement adaptation occurs during precision pull training.**
**a** Illustration of retrograde transduction strategy. **b** Illustration of head-fixed behavior during imaging. **c** Representation of reward zones in training of adaptive and precision isometric pull. **d** Success rate on the isometric pull tasks at three distinct training phases. Mice were proficient on the adaptive isometric pull early in training, whereas only half the mice were able to learn the precision isometric pull task (repeated measures ANOVA, $P = 0.0198$, $F_{2,8} = 6.73$, *Bonferroni post-hoc $P < 0.05$), with the remainder classified as non-learners. **e** Mean pull force per animal across training. **f** Average pull-to-pull correlation coefficients are significantly

higher after learning in precision mice (repeated measures ANOVA, $P = 0.0065$, $F_{1,5} = 22.13$, *Bonferroni post-hoc $P < 0.05$). All data are represented as mean ± sem. **g**–**i** Peak force distributions per trial on isometric pull. Individual data point represents a single trial, unique markers per mouse, and lines connect mean ± sem from each individual. Reward zone is indicated by gray box, trial initiation at 3 g force (dashed line), and max threshold for the adaptive pull is 20 g (top dashed line). The graphs represent results from $n = 5$ mice per group. Source data are provided as a Source Data file.

For imprecise motor learning, we trained mice on the MotoTrak adaptive, isometric pull module. In adaptive pull, the mouse exerted a force exceeding 3 g to initiate a trial and, after 10 trials, a new threshold was set to the 50th percentile of those preceding 10 trials, not exceeding 20 g. Thus, the adaptive threshold is constantly adjusting to the pull force of the animal and does not require modulation of their pull force to a specific range or precise force adjustments for reward. Following behavioral shaping, mice were proficient on the adaptive pull during the first days of learning and maintained this success rate throughout a training period of $16.2 \pm 1.5$ days, a period that was divided into three phases: early, middle, and late (Fig. 1d).

Precision pull involves the same forelimb movements as adaptive pull but requires precise adjustments to restrict pull strength within a specific range of force. The behavior was first shaped in response to mice exerting an isometric pull force through a series of increasingly reduced reward windows, until mice reached the training window of 15–18 g (Supplementary Table 1). Mice were then trained to pull with a peak isometric force within this narrow window over a period of $22.8 \pm 4.0$ days. Successful retrieval significantly increased over training with a slight rise in mean trial pull force (Fig. 1d, e). Moreover, the trained pull movement became significantly more consistent with learning as the correlation of the dynamics of each pull movement to that preceding it (pull-to-pull correlation) increased (Fig. 1f). Over training, there was a shift in the peak force of individual pulls with more than a 1.5-fold increase in success rate, resulting in 51.3% of all pull attempts falling in the reward window and fewer attempts below the lower reward threshold (Fig. 1h, Supplementary Table 2). This shift resulted from an increased accuracy of pulls, rather than reduced variability in peak pull force within a training session (Supplementary Fig. 1a).

While both the precision and adaptive tasks produce the same gross motor movement, learning of the precision pull movement requires active refinement and likely involves adaptive modulation of cortical output. On the precision pull, half the mice failed to improve by at least 25% total success rate over their baseline performance and were categorized as non-learners (Fig. 1d). Non-learners trained for an average of $22.8 \pm 4.4$ days, similar to those that learned the task. Furthermore, in the non-learners neither force nor pull-to-pull correlation increased across training (Fig. 1e, f). Pull dynamics in non-learners were slightly more consistent during the early phase than in mice that successfully learned the precision task. Pull-to-pull correlation values were consistent across training in non-learners, indicating that animals may have failed to explore varied strategies for task success.

## Corticospinal neuron recruitment during precision learning

To observe corticospinal activity during the learning of precise and imprecise isometric pull, we used retrograde transduction at spinal levels C7/8 (segments controlling distal forelimb musculature) to express the genetically encoded calcium indicator GCaMP7f specifically in corticospinal neurons (Fig. 1a). Following transduction, two-photon microscopy was used to record calcium transients from corticospinal somata in layer 5b. To determine the population of labeled corticospinal neurons associated with the trained behavior, we segmented session calcium activity by individual pull movements and averaged activity across successful trials. We set a cutoff threshold of a 20% change in fluorescence above average fluorescence values during movements to determine which neurons were associated with successful isometric pulls. Only $10.2 \pm 1.8\%$ of corticospinal neurons were associated with successful adaptive pulls across learning, while $41.7 \pm 1.1\%$ showed average activity associated with successful precision pulls (Fig. 2b). These associated neurons were then sorted by latency to peak activation (Fig. 2a). Corticospinal neurons associated with successful precision pulls exhibited a pattern of temporal activation when averaged across trials. This temporal activation was not apparent in failed trials, nor in adaptive isometric pull (Fig. 2a). Non-

learners trained on precision pull had $31.2 \pm 0.4\%$ of corticospinal neurons associated with successful pull attempts and a similar pattern of calcium activity on successful attempts (Fig. 2a, b).

We next examined corticospinal activity correlation across training and found that as pull-to-pull correlation increased in animals that learned the precision task, the correlation of corticospinal neuron activity across pulls trended upwards (Fig. 2c). In contrast, mice that failed to learn the precision pull task had exhibited high pull-to-pull correlations throughout training (Fig. 1f) and no overall change in activity correlation (Fig. 2c). On adaptive pull, mice showed a slight increase in pull consistency with training; however, corticospinal activity correlation was largely unchanged. Principal component analysis (PCA) was performed on neuronal activity during individual movement epochs over the initial 50 trials of all training sessions to assess population dynamics across learning. PC1 for individual pulls was projected onto the eigenvector directions to visualize differences across learning and between tasks. Eigenvector directions were defined by loading values calculated using the entirety of the data resulting in similar projection shapes across groups but highlighting shifts in PC scores between groups over learning. Adaptive PC continually shifted over the course of learning while both precision and non-learners converged in the middle to late stages of learning reflecting a refinement of calcium activity that was relatively conserved in the precision tasks (Fig. 2d). Pearson correlation coefficients of individual neuron activity during movement epochs showed a more widespread co-activation of corticospinal neuron pairs in the adaptive pull that increased across training (Fig. 2e), while fewer corticospinal neurons were associated with successful adaptive pulls in late training stage (Fig. 2b). In contrast, precision pull learning was associated with reduced co-activation that was further refined with training (Fig. 2e).

These data suggest that precision learning shapes the activity of relevant corticospinal output networks and that the adaptive task is independent of refinement of these networks. Furthermore, the precision non-learners demonstrated a more limited repertoire of dynamic movements and a corresponding reduction in network modulation, which likely underlies their failure to develop expertise on the precision task.

## Corticospinal dependence of precise behavior

The corticospinal tract is required for dexterous behavior involving precise movements. Transection of the corticospinal tract at the level of the medullary pyramids (pyramidotomy) permanently disrupts dexterous forelimb motor control in rodents and non-human primates[9,11]. After behavioral training, we performed bilateral pyramidotomy to determine the role of corticospinal neurons in the execution of isometric pull tasks (Fig. 3a–c).

First, task acquisition in freely moving animals was similar to that in head-fixed animals: training on adaptive pull showed no effect on success, with peak performance occurring during the initial testing, while mice trained on the precision task showed steadily improved performance throughout training (Fig. 3d). Unlike in head-fixed mice, pull-to-pull movement correlation in freely moving mice showed only a slight increase with precision learning (Fig. 3f). In contrast, mice trained on the adaptive pull showed greater variability in pull-to-pull correlation and peak pull force within trials during early stages of learning; this variability decreased with training (Fig. 3f, Supplementary Fig. 3a). In freely moving mice, the difference in the duration of training required for precision versus adaptive behavior was exacerbated, with training lasting $11.4 \pm 0.2$ days and $34.7 \pm 2.0$ days on the adaptive and precision tasks, respectively. The difference between freely moving and head-fixed isometric pull behavior is not unprecedented, as similar discrepancies have been demonstrated when comparing freely moving and head-fixed single pellet retrieval behavior[23].

We selected the precision range of 15–18 g based upon pilot studies during which all animals could pull well above the target range;

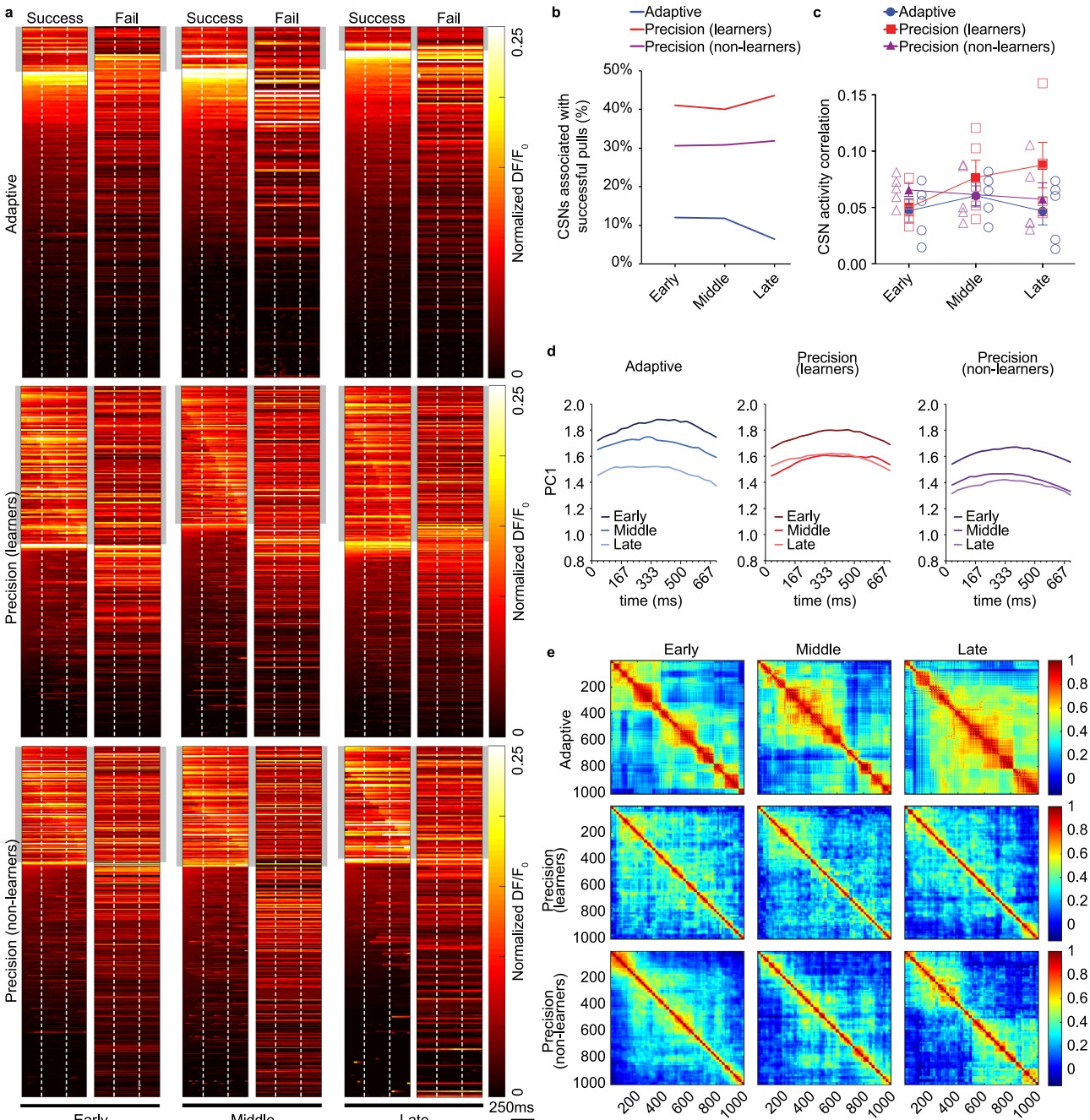

**Fig. 2 | Corticospinal ensembles are selectively active during precision pull training. a** Heat maps show mean activity per neuron normalized across training sessions, separated by successful and failed trials. Movement onset and offset indicated by vertical, dashed, white lines. Gray box indicates cells temporally regulated during pull movements. **b** Percentage of C7/8 corticospinal neurons temporally associated with successful pull trials. **c** Pairwise correlation analysis of activity across movements shows a slight increase with precision pull learning, $n = 5$ mice/group. **d** PCA of corticospinal activity across 50 pulls per training session. PC1 for individual pulls projected onto the eigenvector loadings were visualized to evaluate trends over learning. **e** Pearson correlation coefficients plotted in heatmaps where each point is the correlation coefficient from a pair of neurons. There is more widespread network activation in adaptive pull with pairs of neurons showing higher correlation in activity during movements. Precision coefficients are smaller indicating diverse activity of individual neurons during movement. Data presented as mean ± s.e.m. Source data are provided as a Source Data file.

however, we sought to ensure that performance was independent of animal strength, particularly after pyramidotomy. To do so, we trained a separate group of freely moving mice to pull the isometric handle above a 15 g static threshold. In this static threshold group, mice improved performance across 7.0 ± 1.1 training days (Fig. 3d). Mean peak forces stayed consistent across training and pull-to-pull correlation remained largely unchanged (Fig. 3e, f). These results suggest that the lower success on the precision version of the task was not due to an inability to pull above the lower bound of the force range, but rather the ability to properly modulate the force exerted. This is evident in the standard deviation from the mean as mice trained to pull in the narrow window of the precision task exhibited a standard deviation of 1.20 g, while those trained on the adaptive or static threshold isometric pull had a standard deviation of 4.00 g and 2.77 g, respectively. These imprecise tasks allowed for greater flexibility in strategies for mediating task success.

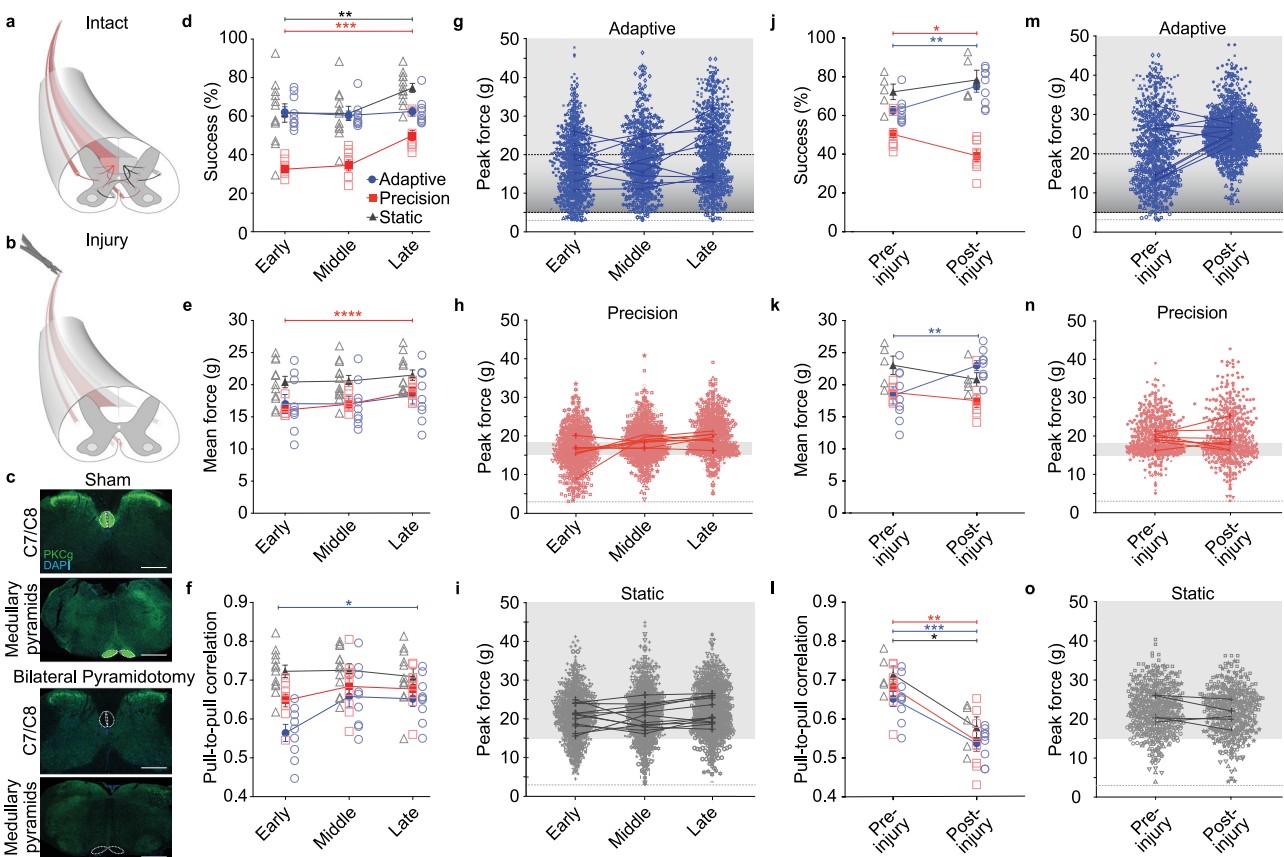

**Fig. 3 | Control of precision isometric pull depends on the corticospinal tract.**
**a**, **b** Illustration of bilateral pyramidotomy of the corticospinal tract.
**c** Representative images of PKCγ staining of the corticospinal tract in sham and bilateral pyramidotomy, below the level of injury. Dashed lines indicate corticospinal tract location, scale bars = 500 μm. **d** Success rate of freely moving mice on the isometric pull task with three thresholds: adaptive, 15 g static, and 15–18 g precision. Performance improves with training on the precision and static conditions (repeated measures ANOVA, precision: $n = 8$ mice, $P < 0.0001$, $F_{1,10} = 41.2$; static: $n = 12$, $P = 0.002$, $F_{2,20} = 9.2$). **e** Mean pull force increases during precision pull training, with limited variability (repeated measures ANOVA, $n = 8$ mice, $P = 0.004$, $F_{1,10} = 12.1$). **f** Pull-to-pull correlation increases across training on adaptive pull (repeated measures ANOVA, $n = 9$, $P = 0.006$, $F_{2,15} = 7.6$, *Bonferroni post-hoc $P < 0.05$). **g–i** Peak force distributions per trial on isometric pull. Individual data point represents a single trial, unique markers per mouse, and lines connect mean ± sem. from each individual. Reward zone is indicated by gray box, trial

initiation at 3 g force (dashed line), and max threshold for adaptive pull is 20 g (top dashed line). **j** Success on precision isometric pull is impaired after bilateral pyramidotomy (two-tailed paired t-test, $n = 8$ mice, $P = 0.010$, df = 7, t = 3.5), while success on the adaptive task is improved after injury (two-tailed paired t-test, $n = 9$, $P = 0.003$, df = 8, t = 4.3). **k** Mean pull force is not significantly affected by pyramidotomy in the precision or static pull. Mice on adaptive pull exhibit a significant increase in mean pull force after injury (two-tailed paired t-test, $n = 9$ mice, $P = 0.001$, df = 8, t = 4.8). **l** Average pull-to-pull correlation coefficients are significantly reduced following injury in all conditions (two-tailed paired t-test, adaptive: $n = 9$, $P = 0.0002$, df = 8, t = 6.5; precision: $n = 8$, $P = 0.0023$, df = 7, t = 4.7; static: $n = 5$, $P = 0.0121$, df = 4, t = 4.4). **m–o** Peak force distributions show more trials outside of reward range only on precision pull after pyramidotomy. Data presented as mean ± sem. in **d–i** for $n = 9$ mice (adaptive), 8 (precision), and 12 (static); and in **j–o** for $n = 9$ (adaptive), 8 (precision), and 5 (static). Source data are provided as a Source Data file.

To determine the dependence of each task on the corticospinal tract, we performed bilateral pyramidotomy after training (Fig. 3a–c). This targeted transection of the corticospinal tract selectively impaired performance on the precision pull task (Fig. 3j). While pull force was not significantly reduced in the absence of the corticospinal tract, animals showed reduced pull-to-pull correlation on all versions of the task after pyramidotomy (Fig. 3k, l). On the precision pull task, this reduced pull correlation can be observed in the increased variability in peak pull forces after injury (Fig. 3n). Despite the increased pull variability, mice trained on the adaptive pull task demonstrated improved performance alongside an increase in pull force after bilateral pyramidotomy (Fig. 3j, k). This improvement may be due to mice simply pulling harder (above 20 g) with more frequency, due to the increased pull-to-pull variability, or perhaps due to a continuation of learning, with mice learning that the adaptive threshold is capped at 20 g of force during the 10.0 ± 0 days of testing following injury. These results indicate that the impairment observed in the precision group is not due to weakness caused by corticospinal injury, but rather by the impairment of movement modulation and lack of viable compensatory movement driven by

spared descending supraspinal circuits. Additionally, the gross movements of the adaptive and static versions of the task are not corticospinal dependent but rather rely on other motor circuits.

## Injury disrupts movement-associated corticospinal activity
We next used bilateral pyramidotomy in combination with in vivo imaging to determine the effects of injury on corticospinal activity during head-fixed isometric pull. As in freely moving animals, bilateral pyramidotomy severely impaired precision isometric pull performance in head-fixed mice (Fig. 4a). In addition, pull-to-pull movement correlation decreased during execution of the precision movement after injury (Fig. 4c). Pyramidotomy resulted in disrupted accuracy of precision pull attempts and most trials in head-fixed mice fell below the reward zone (Fig. 4b, e). The poor performance of the non-learning mice prior to injury was not significantly impacted by pyramidotomy. Non-learning mice showed no significant decrease in pull force or pull-to-pull correlation after injury.

In contrast to precision pull, the execution of adaptive pull was largely unaffected by pyramidotomy. Neither success rate nor pull-to-

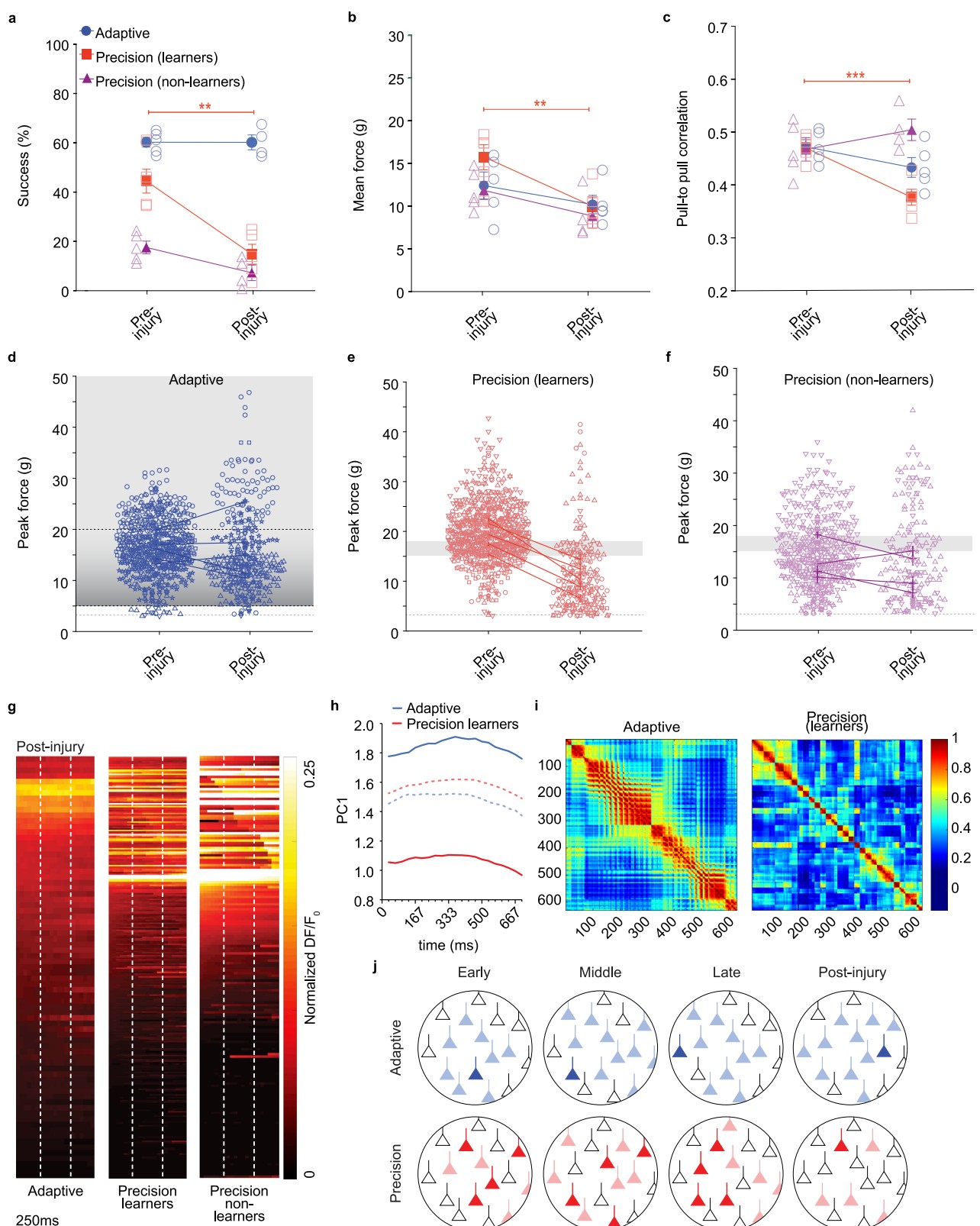

pull correlation were significantly affected by injury (Fig. 4a, c). Unlike in freely moving mice, pull force showed a mild decrease following pyramidotomy. The differences observed between freely moving and head-fixed mice on the precision task is likely due to the more restricted movement of head-fixed animals that could prevent compensatory adaptation of movements. Indeed, freely moving mice demonstrated a greater proportional decrease in pull-to-

pull correlation but a less severe drop-in success rate after injury compared to head-fixed animals.

Following injury, we found a disruption of corticospinal activity during pull movements. On the precision pull task there were few successful trials after injury, therefore we examined the temporal activation of corticospinal neurons across all pulls. Few neurons were associated with pull movement after pyramidotomy across all pulls:

**Fig. 4 | Pyramidotomy disrupts corticospinal activity and precision pull performance. a** In head-fixed mice, the success rate is impaired on precision, but not adaptive isometric pull following bilateral pyramidotomy (two-tailed paired t-test, $P = 0.006$, df = 4, t = 5.3). **b, c** Mean pull force and pull-to-pull correlation on precision pull are significantly impaired by bilateral pyramidotomy (Mean pull force: two-tailed paired t-test, $P = 0.006$, df = 4, t = 5.4; pull-to-pull correlation: two-tailed paired t-test, $P = 0.0002$, df = 4, t = 13.6). **d–f** Peak force distributions show more trials outside of reward range only on precision pull after pyramidotomy. **g** Heat maps show mean activity per neuron normalized across testing sessions. **h** PCA of corticospinal activity across 30 pulls per testing session. PC1 for individual pulls projected onto the eigenvector loadings were visualized to evaluate trends over testing. Dashed lines indicate pre-injury data (late learning) from Fig. 1h. **i** Pearson

correlation coefficients show more widespread network activation in adaptive pull with higher activity correlation in pairs of neurons during movements after pyramidotomy than in precision pull. Precision coefficients decrease following pyramidotomy indicating a disruption of activity with less coactivation post injury. **j** Model of corticospinal network modulation across motor learning and injury. A similar proportion of corticospinal neurons in M1 are active during pull movements in both precision and adaptive tasks across training (shaded neurons); however, a larger proportion of corticospinal neurons are temporally regulated during successful trials (saturated neurons). Pyramidotomy disrupts success-associated circuits. The graphs represent results from $n = 5$ mice per group. Data presented as mean ± sem. Source data are provided as a Source Data file.

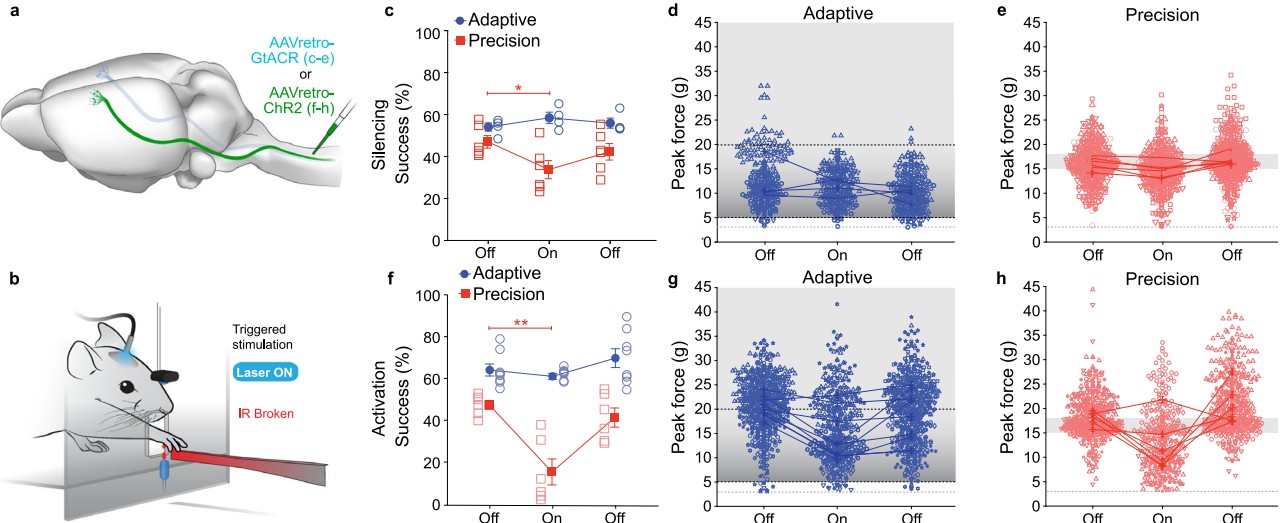

**Fig. 5 | Disruption of corticospinal activity impairs precision motor output. a** Illustration of corticospinal transduction strategy with AAVretro-GtACR or AAVretro-ChR2-tdTomato transduction at C7/8. **b** Illustration of IR-sensor-triggered optogenetic stimulation of M1 with 473 nm fiber coupled laser. **c** GtACR silencing of C7/8 corticospinal neurons (laser On) impairs success on the precision pull (two-tailed paired t-test, $n = 6$ mice, $P = 0.024$, df = 5, t = 3.2). **d, e** Peak pull force distributions of laser On trials show more trials outside the reward zone on

precision pull. **f** C7/8 corticospinal ensemble synchronous activation by ChR2 significantly impairs precision pull success rate (two-tailed paired t-test, $n = 6$ mice, $P = 0.005$, df = 5, t = 4.8). **g, h** Peak pull force distributions show disrupted movements with ChR2 activation of C7/8 corticospinal neurons (laser On), with most precision pull trials outside the reward zone. Data presented as mean ± s.e.m. in **c–e** for $n = 4$ mice (adaptive) and 6 (precision); and in **f–h** for $n = 8$ (adaptive) and 6 (precision). Source data are provided as a Source Data file.

2.2% (adaptive), 22.8% (precision learners), and 28.9% (precision non-learners). Pyramidotomy resulted in a severe decrease in pull-to-pull correlation of precision pull (Fig. 4c). Mice on the adaptive task, and those that failed to learn the precision pull task exhibited little change in pull-to-pull correlation following injury. PCA of corticospinal calcium activity across the initial 30 trials in each testing session showed a large shift in PC scores mapped to eigenvector loadings following injury (Fig. 4h, Supplementary Fig. 4b). Non-learners did not perform enough trials for PCA post-injury. Correlation coefficient analysis indicated a disrupted cortical network with impaired corticospinal activity patterns (Fig. 4i). Pairs of cells within the precision learners displayed less coactivation than the late learning phase while the adaptive group continued to display widespread coactivation post-injury.

## Corticospinal activity regulates precise movements

As pyramidotomy permanently disrupts corticospinal projections, leaving brainstem collaterals intact, we sought to determine the effects of transient disruption of corticospinal neuron activity on dexterous movement. To do so, we selectively expressed *Guillardia theta* anion-conducting channelrhodopsin-2 (GtACR2) in corticospinal neurons projecting to C7/8 spinal cord via retrograde transduction (Fig. 5a). A fiber optic cable was attached to a cannula affixed over primary motor cortex and freely moving mice were trained over a period of

$23.0 ± 2.3$ days and $20.8 ± 0.3$ days for adaptive and precision pull, respectively. Following training, a 473 nm laser triggered by an IR sensor during the reach was used to activate GtACR2 and silence C7/8 corticospinal neurons (Fig. 5b). We observed a subtle, but significant, effect of optogenetic silencing of C7/8 corticospinal neurons. Precision reach success dropped in 5 of 6 mice with confirmed GtACR2 expression in motor cortex (Supplementary Fig. 6), while reach success was unperturbed on the adaptive task (Fig. 5c).

In addition to silencing C7/8 corticospinal neurons during isometric pull, we selectively transduced C7/8 corticospinal neurons to express channelrhodopsin-2 (ChR2) so that we could disrupt the patterned firing of corticospinal ensembles. As above, a fiber optic cable was attached to a cannula affixed over primary motor cortex and freely moving mice were trained to proficiency on the adaptive or precision isometric pull task over $6.5 ± 0.61$ or $18 ± 1.2$ days, respectively. Following training, ChR2 was activated during the reach to disrupt the refined corticospinal network ensembles necessary for successful precision pull. Synchronous optogenetic stimulation of C7/8 corticospinal neurons significantly impaired precision isometric pull performance and reduced mean pull force (Fig. 5f, Supplementary Fig. 5e). While aberrant corticospinal activation did not alter precision pull-to-pull movement correlation, it did result in most pulls falling outside of the reward zone (Fig. 5h, Supplementary Fig. 5f). In contrast to the effects of corticospinal activation on execution of the precision

behavior, mice trained to pull on the imprecise, adaptive isometric pull showed no significant reduction in success rate though there was a significant decrease in pull force during adaptive isometric pull, consistent with a role for the corticospinal neurons modulating force (Fig. 5g, Supplementary Fig. 5e). Mice exhibited no significant reduction in pull-to-pull correlation when performing the adaptive isometric pull during IR-triggered laser ON sessions (Supplementary Fig. 5f). These results confirm that the patterned corticospinal activation observed via in vivo imaging is critical for execution of precise, but not imprecise movements.

## Discussion

The motor cortex is a critical regulator of voluntary movement. In rodents, motor cortex plays an instructional role in the learning of trained behaviors. Studies using learned lever-press tasks have demonstrated that motor cortex is essential in the training stages of the behavior, but becomes disengaged from movements with training and dispensable in movement execution[14,15]. In contrast, complex forelimb reaching movements have shown a greater dependence upon continued motor cortex involvement in trained behavior, with silencing actively disrupting reaching movements[24]. Additionally, sensory-motor integration is critical for adjusting trained behavior to correct movements in response to visual or proprioceptive sensory disruptions[18,25].

In this study, we investigated the role of corticospinal neuron activity in motor cortex during learning of either a precise (precision isometric pull) or imprecise (adaptive isometric pull) task. Training of the complex forelimb movements of the corticospinal-dependent, single pellet reach task has been shown to drive large-scale remodeling of evoked motor output maps in the primary motor cortex (M1), while map changes were not detected after training on a simple lever-press task[7]. Using in vivo imaging of calcium transients, training on a lever-press task was previously found to drive the refinement of excitatory layer 2/3 motor networks[21], but not of corticospinal ensembles[26]. Rather, reduced corticospinal activity correlation was observed alongside reduced movement correlation in late training phases of the lever-press[26]. Similarly, in training of the adaptive isometric pull, we observed no increased correlation of corticospinal activity. There was a slight increase in corticospinal activity correlation across animals with the development of precision pull expertise. What we did observe, were distinct responses of corticospinal neurons to precision and adaptive isometric pull training, with temporal activation of corticospinal ensembles only present during the precision task. Furthermore, PCA showed differing trends at all time points between the adaptive and precision groups indicating divergent patterns of activity during movement epochs throughout learning. In a head-fixed version of the single pellet reach task, layer 5 pyramidal tract neurons have been shown to be similarly time locked to movement[27]. Furthermore, the initial state activity of pyramidal tract neurons is indicative of prior reach outcome, indicating that reinforcement signals in M1 are likely to be driven by outcome signals[27].

A subset of our head-fixed mice failed to learn the precision pull task. Motor training drives declines in movement variability when the reward rate is high as animals attempt to replicate successful trials[28]. Indeed, we found that increased success rate occurred in parallel with increased consistency of trained movements on the precision task. In contrast, mice that exhibited more limited movement variability throughout training failed to learn. PCA showed a divergent response across training sessions between learners and non-learners on the precision pull.

Transection of the corticospinal tract at the level of the medullary pyramids disrupted ensemble activity and selectively impaired performance of precision isometric pull, in line with earlier experimental findings that transection of rodent corticospinal tract abolishes the ability to perform precise dexterous movements[9,29,30]. Transection

resulted in greater impairment in performance on the precision task in head-fixed mice than in freely moving, likely owing to the more restrictive environment of head-fixed animals and the inability for them to make compensatory adjustments to their stance and movement. These compensatory movements may be reflected in the drop in pull-to-pull correlation following injury that occurs only in freely moving mice, though the correlation values are still higher after injury than in the head-fixed mice.

Similar to the effects of pyramidotomy, silencing the C7/8 projecting corticospinal neurons with GtACR after training disrupted pull forces, resulting in a small, but significant deficit in the performance of freely moving mice. It has been reported that there is a decoupling of cortical activity from the execution of highly trained movements, as the basal ganglia begin to play a larger role in the execution of trained movements[31]. Our results demonstrate that corticospinal neurons still play a critical role in the execution of a dexterous movement; however, it may be that this role is reduced following extensive training. If that is the case, we would expect that silencing corticospinal neurons during the early or middle stages of precision pull learning would have a more striking effect on success rate, similar to the effect of silencing M1[15].

In contrast to the effects of silencing or injury, we observed that aberrantly disrupting the temporal activity of C7/8 corticospinal ensembles through synchronized activation of ChR2 fully altered the trained movement and elicited a much more dramatic effect on pull dynamics and task success rate. These neurons send extensive collateral projections to motor centers throughout the brain and brainstem[32–34], so aberrant corticospinal activation is likely to drive disruption of temporal activity in multiple loci regulating the pull movement. The selective disruption of precision pull success rate by widespread corticospinal ensemble activation is in line with the recordings of calcium transients. Corticospinal neurons showed extensive activation during the adaptive isometric pull, which increased across training. In contrast, corticospinal activity during precision pull was temporally regulated with reduced co-activated neurons across training.

Both precision and adaptive isometric pull tasks utilize the same forelimb movements and are therefore likely to engage similar motor circuits; however, only successful execution of the precision task was found to depend on the integrity of the corticospinal tract. The development of expertise occurred rapidly on imprecise tasks and showed no significant correlation between corticospinal activity and movement kinetics; whereas precision expertise required a training-mediated refinement of movement kinetics and temporally regulated corticospinal neuron activity. These findings illustrate a key role in the modulation of corticospinal networks in the learning and maintenance of dexterous movements and provide a powerful toolset for assessing how compensatory mechanisms and corticospinal circuit plasticity can shape recovery of neurological injury.

## Methods

### Animals

All animal work was approved by the Weill Cornell Medicine Institutional Animal Care and Use Committee. C57BL6/J *Rosa26-LSL-tdTomato* and C57BL/6 J mice (Jackson Laboratory) were housed in disposable plastic cages on a reverse 12 h light cycle, humidity 39–48%, and temperature of 21.7 °C. Experiments were conducted on adult male (19.4 ± 0.56 g) and female (16.5 ± 0.42 g) mice during the dark period.

### Surgery

**Spinal cord transduction.** Mice ($n = 29$, 5.83 ± 0.35 weeks) were deeply anesthetized with isoflurane (5% for induction, 1–2% for surgery) heart rate was monitored, and body temperature controlled using SomnoSuite small animal anesthesia system (Kent Scientific). Subcutaneous injection of the analgesic buprenorphine (0.1 mg/kg)

and meloxicam (2 mg/kg) was given immediately after the animal was anesthetized. The fur covering the skin on the dorsum over the cervical spinal cord was shaved and the mouse was secured in a stereotaxic (Stoelting Co.). Sterile eye lubricant was applied to both eyes to prevent corneal drying during surgery and, 0.1 ml of local anesthetic bupivacaine (0.5% solution) was infiltrated into the intended incision site. The skin was cleaned with three sets of alternating Betadine scrub and 70% alcohol then painted with Betadine solution. A midline incision over the vertebra undergoing laminectomy was made, the skin and dorsal musculature incised and retracted to expose the lamina, and partial laminectomy was performed on vertebral body C6 exposing spinal cord segment C7/C8. To minimize spinal motion due to the breathing, we secured the vertebral column at the spinous process of the second thoracic vertebra (T2). The T2 process was slightly elevated and held with spinal cord clamps (Narishige International) to allow exposure of the vertebral interspace and facilitate intraspinal injections. For in vivo imaging: Two 300 nl injections of AAV2retro-pkg-Cre (Addgene 24593, $1.4 \times 10^{13}$ GC/ml) and AAV2retro-syn-jGCaMP7f-WPRE (Addgene 104488, $1 \times 10^{13}$ GC/ml) [1:1 mixture], for optogenetic control: Two 400 nl injection of AAV2retro-GAG-ChR2 (Addgene 28017, $7 \times 10^{12}$ VG/ml) or AAV2retro-CaMKIIa-stGtACR2-FusionRed (Addgene 105669, $7 \times 10^{12}$ VG/ml) were made unilaterally into right spinal cord gray matter at C7/C8 (0.5 mm lateral, 0.5–0.8 mm ventral) were made using a pulled glass pipette with a tip diameter of 35 μm and a Nanoject III programmable injector (Drummond) at a speed of 1 nl/s. The pipet was kept in place for additional 4 min to prevent backflow of the solution. The double layer of dorsal musculature was closed with 7-0 vicryl absorbable surgical suture (Ethicon) and the skin was closed with 7 mm surgical wound clips (Fine Science Tools). Mice were given 1 ml saline (0.9% solution) subcutaneously after surgery and buprenorphine twice daily for 3 days post-operatively.

**Cranial window implantation.** Two weeks after spinal cord transduction, the mice were anesthetized and prepared for surgery, utilizing the technique detailed in the spinal transduction section. Additionally, mice received subcutaneously a single dose of dexamethasone (0.1 mg/kg) to prevent swelling of the brain. The head was placed in a stereotaxic device for stabilization and secured with ear bars and nose restrainer. To remove the skin over the skull, an incision was made at the base of the skull followed by two oblique cuts that converge to the midline. A craniotomy of ~3–4 mm was performed over the motor cortex by thinning the skull with a dental drill. Saline was used while drilling to keep the skull moist and help removal of the bone with minimizing bleeding of the dura. A sterile 5 mm glass coverslip was permanently affixed on top of the dura by applying a cyanoacrylate-based adhesive gel (Loctite 454) around the glass. After the glass was secured, a rectangular aluminum head bar (Narishige International) was affixed to the skull with C&B-Metabond dental adhesive cement.

**Cannula implantation.** During the spinal cord transduction surgery, the mice additionally received subcutaneously a single dose of dexamethasone (0.1 mg/kg) to prevent swelling of the brain. The head was placed in a stereotaxic device for stabilization and secured with ear bars and nose restrainer. To remove the skin over the skull, an incision was made at the base of the skull followed by two oblique cuts that converge to the midline. A craniotomy of 3 mm was performed over the forelimb motor cortex area (AP = 0.5, ML = 1.7 mm) by thinning the skull with a dental drill. Saline was used while drilling to keep the skull moist and help removal of the bone with minimizing bleeding of the dura. A 2.5 mm or 1.25 stainless steel cannula with guide (Thorlabs, CFM22L02) was implanted using Vetbond cyanoacrylate glue (3 M) and C&B-Metabond dental adhesive cement (Parkell).

**Corticospinal tract lesion—bilateral pyramidotomy.** Mice were deeply anaesthetized with isoflurane; an incision of 1 cm was made over the ventral midline. Gently we pushed aside the adipose tissue, we used the manubrium of sternum as a reliable landmark to locate the pyramids ventrally. A small incision was made through the dura with an insulin syringe 28 gauge to facilitate the lesion of the pyramidal tract rostral to atlas, bilaterally at a depth of 0.5 mm with a 15° ophthalmic microscalpel. The skin was closed with 5–0 nylon surgical suture. The sham group underwent a similar surgical procedure without transection of the pyramidal tracts.

**Tissue processing**
Mice were deeply anesthetized with ketamine/xylazine cocktail, transcardially perfused with 4% paraformaldehyde in phosphate buffered saline. Spinal cord and brain were dissected and cryopreserved in 30% sucrose and cryosectioned. Free floating sections (40 μm) were incubated in 0.3% bovine serum albumin in 0.3% Triton X-100 for 30 min, then incubated overnight at 4 °C with primary antibody. The next day, sections were washed and incubated with Alexa Fluor conjugated secondary antibody (1:250; Invitrogen) for 3 h at room temperature then, washed and, sections cover-slipped in Fluoroshield with DAPI (Millipore). Antibodies used for fluorescent immunohistochemistry were: Rabbit anti-PKCγ (1:100; Santa Cruz) and rabbit anti RFP (1:100; Rockland). PKCγ immunoreactivity was used to assess the completeness of the corticospinal transection at the medullary pyramids. RFP was used to label corticospinal neurons in the optogenetic inhibition experiment.

**Water restriction**
C57BL/6 J and Ai14 *LSL-tdTomato* (C57BL/6 J background) mice (*n* = 63) were induced to water restriction over a course of 10 days. Briefly, mice were taken off *ad lib* water and given 2 ml water per day for the first 3 days, 1.5 ml water for the subsequent 3 days, and finally 1 ml water for the final 4 days. Mice were maintained at or above 80% pre-restricted weight and held at 1 ml water per day for the duration of the experiment.

**Isometric pull task**
The MotoTrak behavioral system (Vulintus, Inc.) was used to assess only the right forepaw function. Mice were habituated for 3 days in either the MotoTrak conditions: acrylic chamber for freely moving or head-fixed on the restraint device (MAG-1, Narishige). In both conditions, the pull behavior was first shaped assisting mice to reach the isometric pull with water reward association. Head-fixed mice were divided into two groups: precision and adaptive. In precision isometric pull behavior training, the water reward was first given for a peak isometric pull force between 5 and 20 g (≥75 trials with success of 50% for three days within one week) then, the range was gradually reduced to pull with a force between 13–19 g (≥50 trials) and then to 15–18 g (≥50 trials). Graduation for each training phase required an improvement of at least 25% from their baseline, otherwise, the mice were considered as non-learners. Motor learning in the narrow range of 15–18 g was divided in three phases: early (first 3 days), middle (15.0 ± 4.3 days later) and late (11.6 ± 3.2 days after middle). In adaptive isometric pull behavior training, the adaptive threshold was set to the 50th percentile of the prior 10 trials, starting at 3 g. Additionally, the head-fixed mice underwent imaging of neuron activity in 2 to 4 fields of view over the forelimb motor cortex during a recording session of 6 min. The freely moving mice were divided into three groups: static, precision, and adaptive. The precision isometric pull and adaptive training were similar as described in the head-fixed mice. In static isometric pull behavior training, water reward was first given for an isometric pull force above 5 g. The threshold was increased stepwise to 8 g, 13 g, and finally 15 g.

## Optogenetic experiments

For optogenetic stimulation and silencing of corticospinal neurons, mice ($n = 18$, $6.67 \pm 0.86$ weeks, ChR2; $n = 14$, $8.2 \pm 0.74$ weeks, GtACR2) were trained on the precision isometric pull or adaptive freely moving task, with tethered fiber optic cables. After reaching success rate improvements of at least 25% over baseline for three consecutive days, mice were tested with IR-triggered optogenetic stimulation. As mice reached through the MotoTrak slot to the manipulandum, interruption of an IR beam (Fig. 5b) triggered an Arduino-controlled 473 nm fiber-coupled laser (20 mW). ChR2 activation of corticospinal neurons was performed over at least 50 trials within 15-minute session. Mice were tested again with the laser off the following day. We analyzed success rate, force, pull-to-pull correlation, and pull kinematics. ChR2 expression in corticospinal neurons was visually confirmed through contraction of the right forepaw following optogenetic stimulation in freely moving mice: laser intensity was increased from 0 mW to 50 mW until movement was evoked. In total, 14 of 16 mice showed a response by 20 mW. The remaining 2 mice showed no response, even up to 50 mW stimulation intensity were excluded from the study. An additional 2 mice failed to learn the task and were excluded. GtACR2 expression in corticospinal neurons was confirmed through immunohistochemistry: 4 mice were excluded as 1 mouse failed to learn the task, 1 mouse did not show labeled neurons, and 2 mice experienced postsurgical complications.

## In vivo imaging

A Thorlabs Bergamo 2-photon microscope with SpectraPhysics InsightX3 dual beam laser was used to record calcium dynamics. GCaMP7f signals in corticospinal neurons were recorded at 940 nm using a 16X water-immersion objective (Olympus, WD 3.0 NA 0.8, D32 diameter) in a single imaging plane at depth 650 to 750 mm. All images were acquired at a rate of approximately 30 Hz per frame of $515 \times 512$ pixels using ThorImageLS 3.2 software with an 8 kHz galvo/resonant scanner speed. ThorSync 3.2 software was used in synchrony to store the frame times, triggers and signals coming into and out of the microscope for an off-line data analysis.

## Movement Analysis

Movement period was identified using peak voltage and width extracted from MotoTrak and ThorSync Software. Starting time was marked when the force exceeded a 3 g threshold and end time was marked as the start time plus the width of the pull. For the precision group, peaks between 15–18 g were labeled as successful. Successful trials in the adaptive group were labeled based on the changing threshold recorded in MotoTrak acquisition software. Peak voltage was extracted in all trials per animal for group analysis over learning and injury. Pull correlation was calculated as the average correlation coefficient between consecutive trials within each session. Kinetics plots were created as an average of continuous force output during all trials in all sessions for each time point. The integral of these plots was calculated to quantify observable changes in force and width. The correlation of patterned activity, and inter-trial variability, were compared to success rate and variability in pull dynamics (force/time).

## Imaging analysis

Images were initially pre-processed in FIJI (ImageJ, NIH) with a Gaussian noise filter (Sigma radius: 2.00). Data analysis was performed in MATLAB (MathWorks) using Constrained Nonnegative Matrix Factorization (CNMF)[35]. First, non-rigid motion correction removed motion artifacts. Then the motion-corrected file was split into patches and run through CNMF, de-noising and de-mixing spatially overlapping signals. Neurons selected by radius and the merging threshold (how close components had to be considered the same). Fluorescent traces and $\Delta F/F_0$ were calculated and deconvolved for each identified component. The time-varying baseline ($F_0$) of each fluorescent trace was estimated by averaging the fluorescence over concatenated portions of the trace with no observed calcium events. This estimation was refined by removing the average fluorescence of the background neuropil structures in each observed spatial field. Traces containing less than 5 distinct calcium events were excluded from all further analysis. $\Delta F/F_0$ data was min-max normalized and synchronized with output from the MotoTrak acquisition software through ThorSync Software.

## Fluorescence analysis

To evaluate activity during movements, $\Delta F/F$ was averaged across all pulls in the session for each animal and plotted on heatmaps with each row corresponding to a single neuron. An initial threshold was applied to filter out cells with an average activity within the lowest 10th percentile of the data. Then, another threshold requiring a 20% percent change between the average fluorescence during all movement epochs and the point of highest fluorescence within the trial average was used to filter out cells that did not have an average peak in their activity during movement. The cells that passed these thresholds were sorted based on the timepoint of maximum activity to create the heatmap curve and those that did not pass were sorted by average activity underneath. The percentage of cells that passed both thresholds was calculated to classify cells as movement related.

## PCA

A concatenated matrix of $\Delta F/F$ during movement epochs was created for each individual reach and cell at each behavioral stage. The early, middle, and late training sessions included 50 trials while the post-injury sessions were limited to 30 trials due to fewer trials after injury. PCA was performed across all trials per cell in training and post-injury using the Python module Scikit-learn[36]. PC1 scores were averaged according to behavior and phase of learning then projected onto the eigenvector loadings to visualize differences in task conditions over learning. Heatmaps of the Pearson correlation were created from the initial matrix of concatenated trials to visualize relationships between activity in pairs of cells.

## Activity correlation

Activity correlation during movement was calculated using pairwise correlation analysis. Activity during all distinct movements was concatenated to create a matrix of temporal activity that is movement specific. Pearson's coefficients were calculated between pairs of concatenated activity and were averaged across pairs and animals.

## Statistical analysis

All statistical analysis was performed with GraphPad Prism software version 9.1.0. Parametric tests were used. Learning phase (time)-related changes within a group were tested with repeated measures ANOVA, and post-hoc Bonferroni correction on early late comparisons. Lesion-related changes within a group were tested with paired two-tailed t-test. Data presented as the mean $\pm$ SEM. $P < 0.05$ was considered statistically significant.

## Reporting summary

Further information on research design is available in the Nature Portfolio Reporting Summary linked to this article.

# Data availability

Source data are provided in this paper. Full datasets generated during this study are available in the G-Node GIN public repository: https://doi.gin.g-node.org/10.12751/g-node.sbrmy3/. Source data are provided in this paper.

## Code availability

MATLAB scripts developed for the imaging data analysis are available in the G-Node GIN public repository: https://doi.org/10.12751/g-node. sbrmy3.

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

## Acknowledgements

We thank C. Schaffer (Cornell University) for training in two-photon microscopy methods; Z Chen and P. Christos for help with statistics. This work was supported by the Burke Foundation, the Craig H. Neilsen Foundation, the New York State Department of Health Spinal Cord Injury Research Board C30844GG, and the National Institutes of Health DP2 NS106663, R01 NS105725, and R03 NS103070 to E.H.

## Author contributions

N.S. and E.H. designed the study; N.S., F.M., and Y.M.L. performed experiments; A.M.S. designed and developed the behavioral task; S.F.A. generated all illustrations; F.M., A.B., and M.S. analyzed imaging data; N.S., F.M., Y.M.L., and E.H. wrote the manuscript; N.S. and F.M. made equal contributions to the study.

## Competing interests

A.M.S. is a co-owner and employee of Vulintus, Inc., which sells products used in data collection for this research. A.M.S. and Vulintus had no editorial authority over the decision to publish. The remaining authors declare no competing interests.
