## [Peer Review File · Nature Communications]

Task-specific modulation of corticospinal neuron activity during motor learning in miceReviewers' comments:

Reviewer #1 (Remarks to the Author):

Overall, this is an intriguing manuscript that describes very interesting findings that are important for the motor control and stroke/spinal cord injury fields. In general, there is a long-standing notion that the M1 corticospinal tract (CST) is important for precise control and individuation in primates. This conclusion has been arrived at based on lesion work. This manuscript uses a host of advanced methods to demonstrate the role of the CST in generating precise, low variability movements, i.e. hallmarks of skilled actions. While there are a number of issues that need to be addressed, the results and findings have the potential to be quite important.

Major.

- Characterization of these tasks as "unskilled" and "skilled" is perhaps problematic. Especially as there seems to be some learning over time for all (1g,h,i). It really seems that the main difference is the precision of the output required. Perhaps the term "precise" is better here. Especially, in figure 2, the CST seems to be about regulation of variability. As shown by the data, the animals can still perform movements (which similar strength) but not as precise.

- Also, was "unskilled" always adaptive? If so, that does not seem unskilled. How much did the value change over a session.

- One challenge with comparison of skilled and unskilled is the time of training. It is possible that with similar time of practice, the variability will also reduce, even when the goal zone is less precise. What was the rationale for different training periods? What about comparing the force distributions (i.e. without taking into account the accuracy zone/success zone) for the skilled vs unskilled for similar days of training?

- Figure 1g-i are hard to interpret, as they seem to be session/animal averages. It would help to show distributions of the forces generated and the relative zone for "success." The plots of success and forces are closely related and thus its more informative if combined. Ideally, this could be shown as clouds of forces with the zone labelled.

- 'Unskilled' in late in EDF 1 - the raw figures show very little activity compared to the skilled. However, the averages seem quite similar. Is this example not representative? Does the average conflate true transients versus gross changes in activity.

- Figure 1i 'activity correlation' is hard to interpret. Also, the dynamic values of this measure seem very small (0.37-0.5). In general, there are N2 comparisons for N neurons; was all of this simply averaged per trial and then correlated? Additional details are important to understand the method. Moreover, this might be better done using PCA or factor analysis to capture population level dynamics and then to link PCs/factors to behavior. Otherwise, the pairwise correlation may include a lot of noise and/or task irrelevant signals.

- On a related note, the traces in EDF 1b show slow ramps of activity. The single neuron response in 1a are more typical. Do these neurons show a more precise activation profile per movement?

- There is no histological evidence shown regarding the pyramidal tract lesions. This is important for validation of these findings.

- Figure 4c/d: It is unclear how success can drop while force/correlation are not that affected. Showing the data as distributions of forces would help. Also there seems to be a bigger effect shown in f vs d? (the y-axis do not match for the two plots)

Minor

- Please cite original Lawrence and Kuyper's paper demonstrating the specific dexterity deficits (with recovery of gross motor and grasping) after PT lesions.

- Could the differences in pull forces for head fixed mice vs free mice be due to compensatory movements? For example, using the trunk? Do they have video analysis for this. At the very least, this should be discussed.

- "While early instructional outputs are 3 critical for task acquisition of both skilled and unskilled behavior, primary motor cortex eventually disengages from task execution of unskilled movements as they can be performed during cortical silencing or even in the complete absence of motor cortex." - This seems way too strong a statement and may not be absolutely true (especially for all species). In the Sci Advances paper, it is still not clear if the task was truly dexterous as it involved pushing a lever. This is distinct from the single pellet task that is often used to measure dexterity. In humans and primates, all movements of the arm and hand are usually lost after a

motor cortex stroke; partial recovery can take weeks to months (e.g. see work of Nudo and colleagues; recent kinematic quantification in Khanna et al., Cell, 2021).

- The use of ChR to assess the role of a pathway seems less than ideal. It is unclear how activation is affecting the downstream structures; also, it is possible that these same neurons project to other structures (such as the striatum). This should be discussed.

- Not all of the figure panels are discussed in the text.

Reviewer #2 (Remarks to the Author):

This study by Serradj and colleagues investigated the recruitment and reorganisation of C7/8 corticospinal neurons (CSNs) during skilled and unskilled motor learning. The skilled version of their behavioural task involves precise pull force modulation, where the mouse is required to displace a lever with the correct force to reach a defined 'zone', whereas the unskilled task requires the mouse to simply depress the lever past a designated threshold. Using 2-photon calcium imaging to observe the activity of C7/C8 projecting CSNs in forelimb motor cortex, they present data to support dynamic adjustments in pull force correlating with refinement of corticospinal activity throughout learning. By disrupting descending corticospinal pathways (through pyramidotomy) and corticospinal activity (through optogenetics), the authors suggest that CSNs are not required for unskilled pull behaviour but are necessary for fine force modulation during skilled pull actions.

The role of CSNs in rodent skilled behaviour has been a matter of debate for some time, so the question being addressed is fundamentally important for our understanding of rodent motor control. In general, the methodological approaches are cutting edge, the data are of high quality, providing new insights into CSN activity across motor learning. We have added a list of suggestions below, that we hope when addressed will strengthen the main conclusions and potential impact of the paper.

Main comments:

1) The classification of neurons is somewhat unintuitive. The study aims to investigate the role of CSNs during learning of a 'skilled' or 'unskilled' lever pull task. Therefore, the main interest lies in the activity of CSNs either 1) during the movement or 2) at any other time. To improve understanding, the authors should classify neurons based on whether they show significant changes in $\Delta F/F_0$ during a peri-movement window (pull active) or at any other time (task active, but not movement active). What is the rationale of having three groupings (pull-active, inter-pull active & indiscriminately active)?

Moreover, the data are baseline and peak normalized but there is insufficient detail in the methods to understand how the analysis has been performed. Which period of the trace is used as 'baseline'? Presumably, the time between pulls is a period of quiet non-movement? If this is the case, then this period would provide a stable baseline from which all movement-related activity can be compared. More methodological details should be provided and the rationale for choosing each method should be discussed.

2) Following the comment above, there are no quantitative analyses to support the statement that skilled training increases inhibition prior to movement onset (page 6, line 12). It is difficult to see how this conclusion can be drawn based on the population averages shown in Ext. Data. Fig. 1. If traces were baselined prior to movement then it would be clear to see positive and negative changes in $\Delta F/F_0$. The authors should explore options to better represent individual (Ext. Data. Fig. 1 panel a is very difficult to interpret given the size and the number of lever pull actions included & y axis should be converted from frames to time) and population data (panel b). In summary, how the authors concluded that "Skilled training resulted in the greatest separation of temporal responses in corticospinal classifications, suggesting an increase in inhibition prior to movement onset." is unclear. This should be shown directly and unambiguously, through better presentation of the data, and with references to specific figure panels in the main text.

3) The authors claim that data shown in Fig. 1J/K support the conclusion that “a larger proportion of pull-active neurons associated with successful trials” but the % changes appear to be quite small. Moreover, it is not clear how many field-of-view were incorporated in the analysis, what the variability was within and across FOVs. These data should be added as supplementary information and statistical tests should be performed on data in Fig. 1k to support the main conclusion.

The authors should also provide the same information for Fig. 3g.

4) The pyramidotomy data do not appear to support the main conclusion. In Fig. 2i & 2k, pyramidotomy in freely moving mice profoundly affects the force and pull-to-pull correlations in skilled and unskilled tasks. The authors state “These results demonstrate that skilled isometric pull is corticospinal tract-dependent. In contrast, success on the unskilled task was unaffected by bilateral pyramidotomy.”. Our interpretation of the results is that pyramidotomy affects the ability of the mouse to exert reproducible force, leading to highly variable trial-to-trial movements, but this doesn't affect unskilled task success because mice still have sufficient force to pull the lever past the designated threshold. The movements presumably still have a high degree of variability within the ‘reward zone’? This data should be shown. In contrast, the skilled task requires highly reproducible pull movements to reach and remain within the smaller ‘reward zone’, which due to the high degree of trial-to-trial variability inevitably means that it fails more often and as such the success rate drops. A simpler conclusion would be that pyramidotomy affects all forelimb movements. The same comments apply to the pyramidotomy data shown in Figure 3. The authors should address this directly with additional data and analysis as it forms one of the main conclusions of the study.

The results obtained using a second perturbation approach in freely moving mice are also somewhat confusing. If the predominant movement-related $\Delta F/F_0$ signature is an increase in activity (Fig. 1) then in what way will ChR2 stimulation ‘perturb’ CSN activity? Did the authors explore stimulation frequencies to find one that did in fact disrupt movement related CSN activity? If so, the data should be provided, otherwise one might predict that ChR2 stimulation would result in an accentuation of the existing movement-related increase in $\Delta F/F_0$. As presented, ChR2 stimulation doesn't have a significant effect on pull force or pull-to-pull correlations (Fig. 4c-e) (no stats on figure or in text), how do we know that ChR2 stimulation results in CSN network perturbation? If there is a significant change, presumably this occurs across both skilled and unskilled? If so, the data do not support the main conclusion. This should be explored and clarified in the text, figures, and legend.

5) The manuscript would benefit from the addition of a more in-depth introduction describing all aspects of the proposed study in the context of the published literature and methods section (for example more details on the behaviours, are they self-initiated or cued, how are inter-pull intervals determined?).

Minor comments:

1) The colour scheme should be changed – as an example, Fig. 1, blue represents the reward zone for unskilled and skilled behaviour, force in unskilled behaviour, and the proportion of pull-active neurons. This can become quite confusing.

2) The pull force examples in each figure appear to be at the very low end of the range. Why is this? Perhaps add representative examples.

3) “Two 300 nl injections 1 of AAV2-retro-pkg-Cre (Addgene 24593, 1.4 x 10¹³ GC/ml) and AAV2 retro-syn-jGCaMP7f-WPRE (Addgene 104488, 1 x 10¹³ GC/ml) [1:1 mixture]” – it is not clear why this virus mixture was used? Why is the retro-cre required?

4) Stats appear to be missing in Fig. 1d and 1k, and Figure 4d.

5) All $\Delta F/F_0$ traces should be plotted with ‘time’ on the x axis, not frames.

- 6) CSN activity correlations are very low, the reason why should be discussed in the text.
- 7) Figure 1 legend, panel K – ‘spikes’ should be replaced with a more appropriate term.
- 8) Figure 2 panels are introduced in an odd order (d, e, c, a, b) with no mention of panels f-h. The text should be amended to introduce the figure panels in the same order as they are shown.
- 9) P10, line 22 – “C7/8 projecting corticospinal...” – add the word neurons?
- 10) P11, line 11-14 – “This indicates a likely role for corticospinal collateral projections..... support limited skilled motor recovery after bilateral pyramidotomy” – this is more a point for discussion, rather than a conclusion supported by the data.

Reviewer #3 (Remarks to the Author):

Review of NCOMMS-21-15289-T

This manuscript, by Serradj and colleagues, aims to address the role of the corticospinal tract in the acquisition of skilled movements. Working in mice, the authors examine the acquisition of both skilled and unskilled versions of an isometric task in which the mice apply force to a manipulandum. The authors report results from four basic measurements in mice performing these tasks: activity measurement in corticospinal neurons via two-photon imaging of GCaMP7 across learning stages, task performance measurement following bilateral transection of the corticospinal tract, GCaMP7 imaging in corticospinals following bilateral tract transection, and performance disturbances induced by channelrhodopsin2-mediated activation of corticospinal neurons. The authors discern changes in corticospinal activity that are particular to the skilled version of the task, as well as performance changes specific to the skilled version follow corticospinal tract transection and channelrhodopsin2 activation. The authors suggest these results link the changes they observe in corticospinal activity across learning to a role in learning skilled movements.

This study does not compare favorably to those typically published in Nature Communications. For comparison, a recent study by Veuthey and colleagues (2020) compares very favorably in terms of the experimental design, the statistical and quantitative rigor in supporting claims, the depth of scholarship, the clarity of argumentation, and the impact of its findings. Below I summarize the major concerns I have regarding the present manuscript.

1. The distinctions between corticospinal activity across learning in skilled and unskilled tasks are not convincing. The illustrations of activity changes across learning in Figure 1k and l do not show pronounced or categorical differences between task types. At the early and middle learning time points, when the performance changes across task types are noticeable (Figure 1d), there is no clear difference in activity as quantified in Figure 1k. The difference shown for late time points is also not particularly large, and significance is not tested. In Figure 1l, the variation in pull-pull correlation and CSN activity correlation are small (see axes scales), and the significance of the changes in the relations between these variables for the three behavior types is not demonstrated. Particularly given the small differences shown here in these panels, it is not clear why whatever differences may exist would be functionally meaningful for skilled learning.
2. On a related note, it is not clear how the activity changes described here go beyond those shown by Peters and colleagues (Nature Neuroscience, 2017), which are not referenced in this regard or discussed.
3. The method used to quantify movement cannot show whether the form of movement differs between skilled and unskilled tasks, or between early and late in learning. Though the force profiles look similar on average, the underlying muscle activity could differ between tasks or between time points. Thus, differences in activity across time points, or in the functional involvement of corticospinal neurons between tasks, may not be due to skill learning per se but differences in the muscle activation patterns being executed. In fact, the form of muscle activity is

known to change as rodents learn new single forelimb skills (see Kargo and Nitz, *J Neurosci*, 2004). Moreover, one prominent theory of motor cortical involvement in skill execution is that motor cortex enables particular patterns of muscle activation (synergies) which cannot be generated by subcortical motor centers, such as those that enable individuated joint movements (Shmuelof and Krakauer, *Neuron*, 2011). Such synergies may be increasingly engaged over time to improve performance of the skilled task here.

4. The novelty of the demonstration that the corticospinal tract is involved in skilled forelimb movements is questionable. Previous studies, for example Lawrence and Kuypers (1968) and several studies from Ian Whishaw's group, have shown that effects of corticospinal tract transection are specific to certain types of movements, including those that require a high degree of dexterity.

5. On a related note, the manuscript is in general not well referenced. Many studies relevant to the role of the motor cortex and the corticospinal tract in particular in motor skill execution are not mentioned, and the discussion is rather brief.

6. The authors claim that "Calcium transients recorded from C7/8 projecting corticospinal showed increased volatility in cell classification after injury" (p. 10, line 22). Yet, the authors later state that changes in classification were similar to those seen between mid and late learning. If a similar amount of time elapses between mid and late learning and between late learning and two weeks post injury, it is not clear that the change in cell classification can be attributed to the injury rather than any learning or changes in task strategy that occur over the latter interval.

7. Channelrhodopsin2 activation of corticospinals will quickly perturb activity in many other cell types, as corticospinals send collaterals that synapse onto many other types both cortically and subcortically. Thus, the changes induced by this activation shown in Figure 4 cannot be directly attributed to corticospinal neurons per se. Effects of inactivating corticospinals could more directly implicate activity in corticospinals as functionally relevant.

8. The authors refer in their Discussion to "the development of consistent movement execution" during skill learning (p. 13, line 11), but the small changes in pull-pull correlation across learning do not seem to clearly reflect such development.

9. Some claims in the manuscript are not supported by quantitative measurements or statistical tests but should be. For example, p. 5, line 22: "In contrast, between the middle and late time points, 67% of neurons changed classification in the skilled group, but only 44% of neurons changed categories in the unskilled group." In this case, the significance of the difference between the quantities mentioned is unclear.

We would like to thank the reviewers of our manuscript, “Refinement of corticospinal neuron activity during skilled motor learning,” (NCOMMS-21-15289-T). In the intervening period since receiving your valuable feedback, we have made significant changes to the manuscript and added requested experiments. We believe that the revised manuscript provides a novel and significant advance in the field of motor control that will be of wide interest. Below we address specific concerns and comments.

Sincerely,
Edmund Hollis

Specific responses to reviewer #1

Overall, this is an intriguing manuscript that describes very interesting findings that are important for the motor control and stroke/spinal cord injury fields. In general, there is a long-standing notion that the M1 corticospinal tract (CST) is important for precise control and individuation in primates. This conclusion has been arrived at based on lesion work. This manuscript uses a host of advanced methods to demonstrate the role of the CST in generating precise, low variability movements, i.e., hallmarks of skilled actions. While there are a number of issues that need to be addressed, the results and findings have the potential to be quite important.

We thank the reviewer for your comments and insightful critique of our manuscript. We have incorporated the reviewers’ feedback, addressed the specific concerns outlined below, and present a revised manuscript for your consideration.

Major remarks

Characterization of these tasks as “unskilled” and “skilled” is perhaps problematic. Especially as there seems to be some learning over time for all (1g, h, i). It really seems that the main difference is the precision of the output required. Perhaps the term “precise” is better here. Especially, in figure 2, the CST seems to be about regulation of variability. As shown by the data, the animals can still perform movements (which similar strength) but not as precise.

We agree that the terms may have a variable meaning and interpretation. Indeed our “unskilled” adaptive isometric pull does drive some level of learning with repetition. The skilled versus unskilled terminology was adopted from the study by Kleim *et al.* (1998) showing differences in cortical reorganization between single pellet reach and a simple lever press. We have clarified our nomenclature and revised the labeling of the tasks to “adaptive pull” and “precision pull” based on the reviewer’s feedback. We hope that this provides greater clarity.

-Also, was “unskilled” always adaptive? If so, that does not seem unskilled. How much did the value change over a session.

In renaming the tasks, we have clarified when the adaptive pull was used and when the static 15 g threshold was used. Both would be considered unskilled, as they lack the

need for precision. Within sessions, the threshold values on the adaptive task increased from a baseline of 5 g for the initial 10 trials to 9.8 ± 0.7 g (early training), 11.8 ± 1.6 g (middle), and 13.8 ± 1.5 g (late).

-One challenge with comparison of skilled and unskilled is the time of training. It is possible that with similar time of practice, the variability will also reduce, even when the goal zone is less precise. What was the rationale for different training periods? What about comparing the force distributions (i.e., without taking into account the accuracy zone/success zone) for the skilled vs unskilled for similar days of training?

The rationale for the different training periods is that the skilled behavior requires shaping to learn the behavior. To address the reviewer's concern, we have now compared variability of peak forces per trial over the first 11 days of behavior regardless of the size of the reward zone (Extended Data Figure 1d). We do not observe differences between groups in variability of peak force per trial.

- Figure 1g-l are hard to interpret, as they seem to be session/animal averages. It would help to show distributions of the forces generated and the relative zone for "success." The plots of success and forces are closely related and thus its more informative if combined. Ideally, this could be shown as clouds of forces with the zone labelled.

We would like to thank the reviewer for the suggestion. We have moved the mean force plots per animal to the extended data and replaced them with scatter plots of peak force per trial (Figure 1 d-f). The peak force plots better demonstrate the number of trials that are rewarded (Extended Data Table 2) as variable pull latency leads to a lower average force in the plots of force over time.

- 'Unskilled' in late in EDF 1 - the raw figures show very little activity compared to the skilled. However, the averages seem quite similar. Is this example not representative? Does the average conflate true transients versus gross changes in activity.

The original image showed a period with fewer trials. Occasionally mice show stretches without trials in both behaviors. Activity is measured during trials, so periods without trials do not affect transient detection.

Figure 1i 'activity correlation' is hard to interpret. Also, the dynamic values of this measure seem very small (0.37-0.5). In general, there are N2 comparisons for N neurons; was all of this simply averaged per trial and then correlated? Additional details are important to understand the method. Moreover, this might be better done using PCA or factor analysis to capture population level dynamics and then to link PCs/factors to behavior. Otherwise, the pairwise correlation may include a lot of noise and/or task irrelevant signals.

We would like to thank the reviewer for the suggested approach. The original presentation of activity correlation was averaged per animal so that we could make a direct comparison to the behavioral readout. We have replaced this now with PCA on

neuronal activity during individual movement epochs and correlation coefficients of the population activity during trials.

- On a related note, the traces in EDF 1b show slow ramps of activity. The single neuron response in 1a are more typical. Do these neurons show a more precise activation profile per movement?

Yes, the individual neurons showed a more precise activation profile per movement that was lost with population averaging. The graphs have been removed.

- There is no histological evidence shown regarding the pyramidal tract lesions. This is important for validation of these findings.

Complete transection was confirmed by immunohistochemistry and representative micrographs have been added to Figure 2c.

- Figure 4c/d: It is unclear how success can drop while force/correlation are not that affected. Showing the data as distributions of forces would help. Also there seems to be a bigger effect shown in f vs d? (the y-axis do not match for the two plots)

We have added scatter plots of peak forces per trial to Figure 4 that better show the distribution across trials. Additionally, we have added data from experiments with silencing of C7/8 corticospinal neurons using GtACR2. The mean of the peak pull forces in the original figure had different y-axis values than the plot of force over time as the peak pull force per trial did not always occur at the same time after trial onset, thereby resulting in a lower y-value value in the plots of mean force over time.

Minor remarks

- Please cite original Lawrence and Kuyper's paper demonstrating the specific dexterity deficits (with recovery of gross motor and grasping) after PT lesions.

The citation has been added.

- Could the differences in pull forces for head fixed mice vs free mice be due to compensatory movements? For example, using the trunk? Do they have video analysis for this. At the very least, this should be discussed.

The difference could be attributed to compensatory movements. We do not have video analysis of the postural effects on behavior but have added this point to the discussion.

- "While early instructional outputs are 3 critical for task acquisition of both skilled and unskilled behavior, primary motor cortex eventually disengages from task execution of unskilled movements as they can be performed during cortical silencing or even in the complete absence of motor cortex." - This seems way too strong a statement and may not be absolutely true (especially for all species). In the Sci Advances paper, it is still not clear if the task was truly dexterous as it involved pushing a lever. This is distinct from the single pellet task that is often used to measure dexterity. In humans and primates, all movements of the arm and hand are usually lost after a motor cortex stroke; partial recovery can take

weeks to months (e.g., see work of Nudo and colleagues; recent kinematic quantification in Khanna et al., Cell, 2021).

We agree and have revised the discussion accordingly, with a specific emphasis on the role of CST in the rodent.

- The use of ChR to assess the role of a pathway seems less than ideal. It is unclear how activation is affecting the downstream structures; also, it is possible that these same neurons project to other structures (such as the striatum). This should be discussed.

While the ChR2 results demonstrate that appropriately patterned corticospinal activity is required for precision movement, we agree that it is not specifically due to select connections with C7/8 spinal cord. Indeed, the activation of corticospinal neurons with ChR2 will disrupt the patterned output of corticospinal neurons throughout the motor system, including to the rubrospinal and medullary reticulospinal neurons (Esposito et al., 2014, Mosberger et al., 2017). We have now completed an experiment silencing the C7/8 corticospinal neurons during freely moving isometric pull and found a decline in precision performance similar to that following pyramidotomy in freely moving animals (Figure 4c-e).

- Not all of the figure panels are discussed in the text.

We have edited the text accordingly.

Specific responses to reviewer #2

This study by Serradj and colleagues investigated the recruitment and reorganization of C7/8 corticospinal neurons (CSNs) during skilled and unskilled motor learning. The skilled version of their behavioural task involves precise pull force modulation, where the mouse is required to displace a lever with the correct force to reach a defined 'zone', whereas the unskilled task requires the mouse to simply depress the lever past a designated threshold. Using 2-photon calcium imaging to observe the activity of C7/C8 projection CSNs in forelimb motor cortex, they present data to support dynamic adjustments in pull force correlating with refinement of corticospinal activity throughout learning. By disrupting descending corticospinal pathways (through pyramidotomy) and corticospinal activity (through optogenetics), the authors suggest that CSNs are not required for unskilled pull behaviour but are necessary for fine force modulation during skilled pull actions.

The role of CSNs in rodent skilled behaviour has been a matter of debate for some time, so the question being addressed is fundamentally important for our understanding of rodent motor control. In general, the methodological approaches are cutting edge, the data are of high quality, providing new insights

into CSN activity across motor learning. We have added a list of suggestions below, that we hope when addressed will strengthen the main conclusions and potential impact of the paper.

We thank the reviewer for your comments and insightful critique of our manuscript. We have incorporated your feedback, addressed the specific concerns outlined below, and present a revised manuscript for your consideration.

Main comments:

1) The classification of neurons is somewhat unintuitive. The study aims to investigate the role of CSNs during learning of a 'skilled' or 'unskilled' lever pull task. Therefore, the main interest lies in the activity of CSNs either 1) during the movement or 2) at any other time. To improve understanding, the authors should classify neurons based on whether they show significant changes in $\Delta F/F_0$ during a peri-movement window (pull active) or at any other time (task active, but not movement active). What is the rationale of having three groupings (pull-active, inter-pull active & indiscriminately active)?

Thank you for this comment. In the original iteration of the manuscript, we used the three classifications of Peters *et al.*, 2017, to group corticospinal neurons into pull-active, inter-pull interval active, and indiscriminately active. As this may result in classifying some neurons specific to other movements that occur in between pulls, we have edited as suggested to classify neurons as movement active in the peri-movement window. From activity averaged across successful trials, corticospinal neurons with greater than a 20% change in fluorescence above average values during movements were classified as movement active.

Moreover, the data are baseline and peak normalized but there is insufficient detail in the methods to understand how the analysis has been performed. Which period of the trace is used as 'baseline'? Presumably, the time between pulls is a period of quiet non-movement? If this is the case, then this period would provide a stable baseline from which all movement-related activity can be compared. More methodological details should be provided and the rationale for choosing each method should be discussed.

We have clarified and expanded the methods. The time-varying baseline (F_0) of each fluorescent trace was estimated by averaging the fluorescence over concatenated portions of the trace with no observed calcium events. This estimation was refined by removing the average fluorescence of the background neuropil structures in each observed spatial field.

2) Following the comment above, there are no quantitative analyses to support the statement that skilled training increases inhibition prior to movement onset (page 6, line 12). It is difficult to see how this conclusion can be drawn based on the population averages shown in Ext. Data. Fig. 1. If traces were baselined prior

to movement then it would be clear to see positive and negative changes in $\Delta F/F_0$. The authors should explore options to better represent individual (Ext. Data. Fig. 1 panel a is very difficult to interpret given the size and the number of lever pull actions included & y axis should be converted from frames to time) and population data (panel b). In summary, how the authors concluded that “Skilled training resulted in the greatest separation of temporal responses in corticospinal classifications, suggesting an increase in inhibition prior to movement onset.” is unclear. This should be shown directly and unambiguously, through better presentation of the data, and with references to specific figure panels in the main text.

The statement has been removed and we have revised the analysis of GCaMP activity presented in figure 1. We have added PCA on neuronal activity during individual movement epochs and correlation coefficients of the population activity during trials.

3) The authors claim that data shown in Fig. 1J/K support the conclusion that “a larger proportion of pull-active neurons associated with successful trials” but the % changes appear to be quite small. Moreover, it is not clear how many field-of-view were incorporated in the analysis what the variability was within and across FOVs. These data should be added as supplementary information and statistical tests should be performed on data in Fig. 1k to support the main conclusion. The authors should also provide the same information for Fig. 3g. We have significantly revised figures 1 and 3. Furthermore, we’ve now included full information on FOVs per time point in Extended Data Table 4.

4) The pyramidotomy data do not appear to support the main conclusion. In Fig. 2i & 2k, pyramidotomy in freely moving mice profoundly affects the force and pull-to-pull correlations in skilled and unskilled tasks. The authors state “These results demonstrate that skilled isometric pull is corticospinal tract-dependent. In contrast, success on the unskilled task was unaffected by bilateral pyramidotomy.”. Our interpretation of the results is that pyramidotomy affects the ability of the mouse to exert reproducible force, leading to highly variable trial-to-trial movements, but this doesn’t affect unskilled task success because mice still have sufficient force to pull the lever past the designated threshold. The movements presumably still have a high degree of variability within the ‘reward zone’? This data should be shown. In contrast, the skilled task requires highly reproducible pull movements to reach and remain within the smaller ‘reward zone’, which due to the high degree of trial-to-trial variability inevitably means that it fails more often and as such the success rate drops. A simpler conclusion would be that pyramidotomy affects all forelimb movements. The same comments apply to the pyramidotomy data shown in Figure 3. The authors should address this directly with additional data and analysis as it forms one of the main conclusions of the study.

We have revised the data analysis and discussion. We have now included the peak force distributions per trial (Figure 2m-o), which demonstrate that pyramidotomy disrupts the accuracy of precision pulls. We have also added a discussion on the differences between freely moving and head-fixed animals as we do observe changes in pull-to-pull correlation of adaptive and precision isometric pull in freely moving but not head-fixed animals after bilateral pyramidotomy. This is likely due to changes in latency, rather than force, as there are limited effects on adaptive pull force (Figure 2k,m).

The results obtained using a second perturbation approach in freely moving mice are also somewhat confusing. If the predominant movement-related $\Delta F/F_0$ signature is an increase in activity (Fig. 1) then in what way will ChR2 stimulation 'perturb' CSN activity? Did the authors explore stimulation frequencies to find one that did in fact disrupt movement related CSN activity? If so, the data should be provided, otherwise one might predict that ChR2 stimulation would result in an accentuation of the existing movement-related increase in $\Delta F/F_0$. As presented, ChR2 stimulation doesn't have a significant effect on pull force or pull-to-pull correlations (Fig. 4c-e) (no stats on figure or in text), how do we know that ChR2 stimulation results in CSN network perturbation? If there is a significant change, presumably this occurs across both skilled and unskilled? If so, the data do not support the main conclusion. This should be explored and clarified in the text, figures, and legend.

We have revised extensively. We have now included heat maps for calcium activity showing the temporal regulation of corticospinal neuron activity. The predominant movement-related change in fluorescence was not a general increase during precision isometric pull, but rather temporal regulation and more refined co-activation of corticospinal neurons (Figure 1g,i). We observe more widespread corticospinal network activation on adaptive pull. We have now plotted the peak force distributions across trials (Figure 4h), which makes it easier to observe the effects of synchronous corticospinal activation on the accuracy of precision isometric pull movements. Additionally, we have now added an optogenetic silencing experiment, which results in disrupted movements similar to pyramidotomy in freely moving mice. We have edited the discussion accordingly.

5) The manuscript would benefit from the addition of a more in-depth introduction describing all aspects of the proposed study in the context of the published literature and methods section (for example more details on the behaviours, are they self-initiated or cued, how are inter-pull intervals determined?).

We have extensively edited the manuscript to add more context and greater details.

Minor comments:

1) The colour scheme should be changed – as an example, Fig. 1, blue represents the reward zone for unskilled and skilled behaviour, force in unskilled behaviour, and the proportion of pull-active neurons. This can become quite confusing

We have significantly revised figure 1.

2) The pull force examples in each figure appear to be at the very low end of the range. Why is this? Perhaps add representative examples.

In the original version, we included the mean pull kinetics over time. By averaging over time, across animals, the mean force at a given time point is lower than the peak force.

We have clarified the difference between mean and peak trial force and now present the peak force in the main figures (along with reward zones) for greater clarity. Mean force profiles can be found in Extended Data Figure 1h, 2c, and 4a-e.

3) “Two 300 nl injections 1 of AAV2-retro-pkg-Cre (Addgene 24593, 1.4 x 10¹³ GC/ml) and AAV2 retro-syn-jGCaMP7f-WPRE (Addgene 104488, 1 x 10¹³ GC/ml) [1:1

3 mixture]” – it is not clear why this virus mixture was used? Why is the retro-cre required?

We apologize for the lack of clarity. We used AAV2retro-pgk-Cre transduction in Ai14 *Rosa-LSL-tdTomato* mice in order to visualize corticospinal neurons for use as landmarks (along with cerebral vasculature) to align fields of view across multiple imaging sessions.

4) Stats appear to be missing in Fig. 1d and 1k, and Figure 4d

We have edited the figure legends accordingly. Also, please find a summary of all statistics values in the reporting table, Extended Data Table 5.

5) All $\Delta F/F_0$ traces should be plotted with ‘time’ on the x axis, not frames.

All temporal data is now presented as time.

6) CSN activity correlations are very low, the reason why should be discussed in the text

We have revised the discussion in regard to corticospinal activity. Indeed, the magnitude of corticospinal activity correlation values that we calculated were similar to those observed in earlier studies (Peters *et al.* 2017).

7) Figure 1 legend, panel K – ‘spikes’ should be replaced with a more appropriate term.

The classification of neurons based on CNMF spike estimation has been removed.

8) Figure 2 panels are introduced in an odd order (d, e, c, a, b) with no mention of panels f-h. The text should be amended to introduce the figure panels in the same order as they are shown.

We have revised Figure 2 and updated the text.

9) P10, line 22 – “C7/8 projecting corticospinal....” – add the word neurons?

The text has been edited.

10) P11, line 11-14 – “This indicates a likely role for corticospinal collateral projections..... support limited skilled motor recovery after bilateral pyramidotomy” – this is more a point for discussion, rather than a conclusion supported by the data.

The sentence has been removed.

Specific responses to reviewer #3

Review of NCOMMS-21-15289-T

This manuscript, by Serradj and colleagues, aims to address the role of the corticospinal tract in the acquisition of skilled movements. Working in mice, the authors examine the acquisition of both skilled and unskilled versions of an isometric task in which the mice apply force to a manipulandum. The authors report results from four basic measurements in mice performing these tasks: activity measurement in corticospinal neurons via two-photon imaging of GCaMP7 across learning stages, task performance measurement following bilateral transection of the corticospinal tract, GCaMP7 imaging in corticospinals following bilateral tract transection, and performance disturbances induced by channelrhodopsin2-mediated activation of corticospinal neurons. The authors discern changes in corticospinal activity that are particular to the skilled version of the task, as well as performance changes specific to the skilled version following corticospinal tract transection and channelrhodopsin2 activation. The authors suggest these results link the changes they observe in corticospinal activity across learning to a role in learning skilled movements.

This study does not compare favorably to those typically published in Nature Communications. For comparison, a recent study by Veuthey and colleagues (2020) compares very favorably in terms of the experimental design, the statistical and quantitative rigor in supporting claims, the depth of scholarship, the clarity of argumentation, and the impact of its findings. Below I summarize the major concerns I have regarding the present manuscript.

Thank you for your thoughtful consideration of our manuscript. We have significantly revised the data analysis, presentation, and discussion of our findings in response to your critique as well as the points raised by the other two reviewers. We hope that you find the revised manuscript more suitable.

Major concerns:

1. The distinctions between corticospinal activity across learning in skilled and unskilled tasks are not convincing. The illustrations of activity changes across learning in Figure 1k and I do not show pronounced or categorical differences

between task types. At the early and middle learning time points, when the performance changes across task types are noticeable (Figure 1d), there is no clear difference in activity as quantified in Figure 1k. The difference shown for late time points is also not particularly large, and significance is not tested. In Figure 1l, the variation in pull-pull correlation and CSN activity correlation are small (see axes scales), and the significance of the changes in the relations between these variables for the three behavior types is not demonstrated. Particularly given the small differences shown here in these panels, it is not clear why whatever differences may exist would be functionally meaningful for skilled learning.

We have revised the analysis of corticospinal activity data based upon the prior reviews. Correlation coefficients of the population activity during trials show clear differences in co-activation of neuron pairs, with less specificity during unskilled, adaptive movements than with skilled, precision isometric pull (Figure 1h,i).

2. On a related note, it is not clear how the activity changes described here go beyond those shown by Peters and colleagues (Nature Neuroscience, 2017), which are not referenced in this regard or discussed.

We have added a section to the discussion that puts our findings on skilled versus unskilled motor learning in context with earlier work, including that of Peters *et al.*

3. The method used to quantify movement cannot show whether the form of movement differs between skilled and unskilled tasks, or between early and late in learning. Though the force profiles look similar on average, the underlying muscle activity could differ between tasks or between time points. Thus, differences in activity across time points, or in the functional involvement of corticospinal neurons between tasks, may not be due to skill learning per se but differences in the muscle activation patterns being executed. In fact, the form of muscle activity is known to change as rodents learn new single forelimb skills (see Kargo and Nitz, J Neurosci, 2004). Moreover, one prominent theory of motor cortical involvement in skill execution is that motor cortex enables particular patterns of muscle activation (synergies) which cannot be generated by subcortical motor centers, such as those that enable individuated joint movements (Shmuelof and Krakauer, Neuron, 2011). Such synergies may be increasingly engaged over time to improve performance of the skilled task here.

While the movement needed to pull on the handle is similar across our isometric pull paradigms, it is indeed possible that muscle synergies are changing with the development of expertise in our task. We would posit that a refinement of the command of muscle synergies could be a mechanism to improve accuracy during learning of the precision isometric pull.

4. The novelty of the demonstration that the corticospinal tract is involved in skilled forelimb movements is questionable. Previous studies, for example Lawrence and Kuypers (1968) and several studies from Ian Whishaw's group,

have shown that effects of corticospinal tract transection are specific to certain types of movements, including those that require a high degree of dexterity.

The field of motor control has a rich and deep knowledge base and we have significantly revised our manuscript to put our findings in context. Of course, we do not claim novelty in describing the role of corticospinal neurons in dexterous movements in the rodent but hope to add to the discussion of the role of corticospinal activity during distinct aspects of motor control.

5. On a related note, the manuscript is in general not well referenced. Many studies relevant to the role of the motor cortex and the corticospinal tract in particular in motor skill execution are not mentioned and the discussion is rather brief.

In our original submission we sought to abide by manuscript length constraints and cut critical parts of the discussion. We have significantly revised and expanded the current manuscript to put our findings in context.

6. The authors claim that “Calcium transients recorded from C7/8 projecting corticospinal showed increased volatility in cell classification after injury” (p. 10, line 22). Yet, the authors later state that changes in classification were similar to those seen between mid and late learning. If a similar amount of time elapses between mid and late learning and between late learning and two weeks post injury, it is not clear that the change in cell classification can be attributed to the injury rather than any learning or changes in task strategy that occur over the latter interval.

As the reviewers have pointed out, these classifications provided little value and have been removed.

7. Channelrhodopsin2 activation of corticospinals will quickly perturb activity in many other cell types, as corticospinals send collaterals that synapse onto many other types both cortically and subcortically. Thus, the changes induced by this activation shown in Figure 4 cannot be directly attributed to corticospinal neurons per se. Effects of inactivating corticospinals could more directly implicate activity in corticospinals as functionally relevant.

We agree with the reviewer that the selective activation of C7/8 corticospinal neurons with channelrhodopsin does not specifically disrupt their patterned activity within the spinal cord. Indeed, these neurons make extensive collateral connections throughout the motor circuit. We have revised the discussion of our findings on synchronous activation of corticospinal neurons.

Additionally, we have now performed experiments with silencing of C7/8 corticospinal neurons using the *Guillardia* theta anion-conducting channelrhodopsin2 (GtACR2). Transient silencing of corticospinal neurons during the precision isometric pull movement disrupted the accuracy of pull force in much the same way as pyramidotomy (Figure 4c,e), with the exception of not affecting pull-to-pull correlation within the session (Extended Data Figure 4c). The effects were less robust than optogenetic activation with ChR2, likely owing to the effects of aberrant collateral activation or co-

activation of antagonistic forelimb muscles. Perhaps silencing of corticospinal neurons during earlier stages of the learning process would have a greater effect than silencing in expert animals.

8. The authors refer in their Discussion to “the development of consistent movement execution” during skill learning (p. 13, line 11), but the small changes in pull-pull correlation across learning do not seem to clearly reflect such development.

We have revised the discussion to better reflect our findings and put them in context of the prior art.

9. Some claims in the manuscript are not supported by quantitative measurements or statistical tests but should be. For example, p. 5, line 22: “In contrast, between the middle and late time points, 67% of neurons changed classification in the skilled group, but only 44% of neurons changed categories in the unskilled group.” In this case, the significance of the difference between the quantities mentioned is unclear.

As the reviewers have pointed out, these classifications provided little value and have been removed.

REVIEWER COMMENTS

Reviewer #1 (Remarks to the Author):

The manuscript is much improved. I appreciate the responsiveness. I think this is an important contribution and worthy of publication. Addressing the following issues can improve the manuscript.

I am not sure how to interpret the PCA traces. There seem to be DC offsets and not changes in modulation depth for the PC1 time series. It might be worth seeing if trial averaged data using only fast successful trials are better at estimating activations. Comparison of the variance accounted for (VAF) might help. Plotting VAF curves might show a more complex CST recruitment than for 'adaptive' (i.e., a few PCs explain more variance for that over the precision task).

The terms 'skilled' and 'unskilled' are still confusing as used here. For example, the conflation of adaptive and unskilled seems odd. The difference between adaptive and precise can be confusing over time. Perhaps it could be framed as imprecise or precise movement control (or something else)?

I also find the text in the intro/ discussion about engagement and disengagement somewhat confusing. The intro/discussion links this concept to the lever press task and 'complex' forelimb tasks. There is also link made to prehension vs gross motor. It would help to propose a more comprehensive model including the potential contribution of precision and prehension. Moreover, since the Kuypers study is cited, it would help to have more comparison between primates and rodents. There even untrained (? Unskilled) fine control seems to show long term deficits. The Khanna et al 2021 study also shows fine motor deficits after a lesion even with prior long-term training.

Minor:

I think EDF 1 is helpful and may be better as a main figure.

Reviewer #2 (Remarks to the Author):

The authors should be commended on the thorough revision of the original manuscript. They have now addressed in full all of my previous concerns.

One point to note, in the introduction the knowledge gap or core question doesn't appear until very late and even then it is not explicitly stated. The authors should revise the first intro paragraph to highlight the knowledge gap and question to be addressed.

Reviewer #3 (Remarks to the Author):

Review of NCOMMS-21-15289A-Z (resub)

The authors have substantially revised the previous manuscript and have largely resolved the concerns I raised in my previous review. In particular, the discussion is substantially improved, and relevant literature has been cited. I have several remaining concerns that should be addressed before publication.

Major concerns

The authors still claim that the two movements they train (adaptive and precision) involve the same muscle activation - lines 89 and 361. This is not demonstrated. It may be clear that the movements are similar, but it is not clear that the underlying muscle activation is the same. It is well established in the literature that the learning of new precision movements can involve increased levels of antagonist muscle cocontraction, and so it is very possible that the precision

movement these authors train also involves substantially more cocontraction, and thus substantially different muscle activation patterns compared to the adaptive version. The authors should not imply that muscle activity is equivalent in both of their tasks.

On a related note, the fact that muscle activation could be overall more vigorous for the precision movement because of increased cocontraction could explain why more neurons are significantly task modulated in calcium imaging data, given the manner in which task modulation is determined. Thus it remains possible that the increased activity levels seen in imaging reflect increased muscle activation, rather than the increased involvement of corticospinal neurons in the precision task.

The authors find a reduction in the fraction of task modulated cells following the transection of the corticospinal tract, but given the way they quantify task modulation, and given that the vigor of movement does appear to be reduced in the head-fixed version of the precision task (Fig. 3e), it is not clear that the transection “decoupled” corticospinal neuron from behavior. That claim would seemingly require a movement of similar strength despite reduced activation of corticospinal neurons.

The claim that temporal modulation of firing is only seen during the precision version of the task does not seem to square with the shape of the first principal components, which seem to show similar levels of modulation for both the adaptive and precision versions of the task (Figures 1h and 3h).

Minor concerns:

Line 48: should be “require” not “requires”

Line 58: “enhanced signal-to-noise ratios within primary motor cortex” - this phrase is vague and should be clarified

Line 77: “non-specific” - unclear what this means; suggest removing word

Line 172: “Correlation coefficients from a covariance matrix” - nonstandard language, unclear what is meant here - the values in covariance matrices are not correlation coefficients.

Line 178: It is not clear why the changes described just above (line 173: “widespread coactivation of corticospinal neuron pairs in the adaptive pull”) would not constitute refinement.

Line 292: “These results confirm ...” - this claim is not supported; while it is true that the above results in the previous section on the inactivation experiments do support this claim, the fact that corticospinal neuron activation will in turn activate many other neurons in an aberrant way and disturb behavior does not specifically implicate corticospinal neurons per se in the task.

We would like to thank the reviewers for taking time to once again evaluate our manuscript, now entitled, “Task-specific modulation of corticospinal neuron activity during motor learning,” (NCOMMS-21-15289A-Z). We have incorporated your feedback and revised the manuscript for your further consideration.

Sincerely,
Edmund Hollis

Specific responses to reviewer #1

The manuscript is much improved. I appreciate the responsiveness. I think this is an important contribution and worthy of publication. Addressing the following issues can improve the manuscript.

We thank the reviewer for your consideration and have addressed the specific concerns outlined below.

I am not sure how to interpret the PCA traces. There seem to be DC offsets and not changes in modulation depth for the PC1 time series. It might be worth seeing if trial averaged data using only fast successful trials are better at estimating activations. Comparison of the variance accounted for (VAF) might help. Plotting VAF curves might show a more complex CST recruitment than for ‘adaptive’ (i.e., a few PCs explain more variance for that over the precision task).

To track PCA values across pull movements, PCA traces were plotted as the PC scores mapped onto the eigenvector directions. Since PCA was calculated using all combined groups and time points, the eigenvector loadings remain the same between groups and across time. This method was used to highlight differences between groups during learning as opposed to running separate PCAs for each time point. Since the loadings are the same, there are no changes in modulation depth between groups. Therefore, differences in DC offsets are indicative of differences in PC scores between groups and across learning. Mean PC1 and PC2 values are also plotted in the revised Extended Data Figures 2 and 4.

We included all trials here as there are not enough successful trials post injury to perform this analysis. Additionally, a pull lasts an average of 238 ms with variance on the order of milliseconds, so fast successful trials would be indistinguishable from slow trials using GCaMP7f.

In addition, the VAF curves (right) for PCA performed on each group individually do not show significant variation in the number of PCs required to explain the variance compared to the PCA performed on the complete data set.

The terms ‘skilled’ and ‘unskilled’ are still confusing as used here. For example, the conflation of adaptive and unskilled seems odd. The difference between adaptive and precise can be confusing over time. Perhaps it could be framed as imprecise or precise movement control (or something else)?

We have adjusted the language and adopted the suggested precise and imprecise labeling. We also use dexterous at times to specifically describe the movement.

I also find the text in the intro/ discussion about engagement and disengagement somewhat confusing. The intro/discussion links this concept to the lever press task and ‘complex’ forelimb tasks. There is also link made to prehension vs gross motor. It would help to propose a more comprehensive model including the potential contribution of precision and prehension. Moreover, since the Kuypers study is cited, it would help to have more comparison between primates and rodents. There even untrained (? Unskilled) fine control seems to show long term deficits. The Khanna et al 2021 study also shows fine motor deficits after a lesion even with prior long-term training.

We have revised the discussion on engagement/disengagement to be clearer in regard to the role of motor cortex and corticospinal circuits in fine motor control. Indeed, Lawrence and Kuypers observed permanent deficits in dexterous motor control after pyramidotomy. Stroke models, such as in the Khanna paper, result clear deficits in motor control while perilesional cortex and adjacent structures provide the neural substrate to recover some of the lost function. We’ve sharpened the focus of this section, though plasticity mechanisms after cortical injury are certainly interesting points of consideration for a more in-depth review.

Minor:

I think EDF 1 is helpful and may be better as a main figure.

Thank you for the suggestion. We have included aspects of Extended Data Figure 1 in the main figures and split Figure 1 into two separate figures to accommodate the change.

Specific responses to reviewer #2

The authors should be commended on the thorough revision of the original manuscript. They have now addressed in full all of my previous concerns.

One point to note, in the introduction the knowledge gap or core question doesn’t appear until very late and even then it is not explicitly stated. The authors should revise the first intro paragraph to highlight the knowledge gap and question to be addressed.

We thank the reviewer for considering our manuscript. We have revised the introductory paragraph to better highlight the knowledge gap and question we addressed.

Specific responses to reviewer #3

Review of NCOMMS-21-15289A-Z (resub)

The authors have substantially revised the previous manuscript and have largely resolved the concerns I raised in my previous review. In particular, the discussion is substantially improved, and relevant literature has been cited. I have several remaining concerns that should be addressed before publication.

Thank you for your critical evaluation of our manuscript. We have revised the manuscript in response to your comments.

Major concerns

The authors still claim that the two movements they train (adaptive and precision) involve the same muscle activation - lines 89 and 361. This is not demonstrated. It may be clear that the movements are similar, but it is not clear that the underlying muscle activation is the same. It is well established in the literature that the learning of new precision movements can involve increased levels of antagonist muscle cocontraction, and so it is very possible that the precision movement these authors train also involves substantially more cocontraction, and thus substantially different muscle activation patterns compared to the adaptive version. The authors should not imply that muscle activity is equivalent in both of their tasks.

Thank you for your comment. Yes, we agree that muscle activation patterns may be quite different despite the movements being similar. We have revised the text, removing mention of muscle activation as we do not have EMG recordings at present.

On a related note, the fact that muscle activation could be overall more vigorous for the precision movement because of increased cocontraction could explain why more neurons are significantly task modulated in calcium imaging data, given the manner in which task modulation is determined. Thus it remains possible that the increased activity levels seen in imaging reflect increased muscle activation, rather than the increased involvement of corticospinal neurons in the precision task.

That is an interesting point to consider, though temporal modulation of the corticospinal neurons would still likely be required for differential patterns of task-specific muscle activation.

The authors find a reduction in the fraction of task modulated cells following the transection of the corticospinal tract, but given the way they quantify task modulation, and given that the vigor of movement does appear to be reduced in the head-fixed version of the precision task (Fig. 3e), it is not clear that the transection “decoupled” corticospinal neuron from behavior. That claim would seemingly require a movement of similar strength despite reduced activation of corticospinal neurons.

We have edited the text to better describe our findings.

The claim that temporal modulation of firing is only seen during the precision version of the task does not seem to square with the shape of the first principal components, which seem to show similar levels of modulation for both the adaptive and precision versions of the task (Figures 1h and 3h).

Reviewer number 1 had similar concerns regarding the PCA presentation. As stated above, we plotted PC scores onto the eigenvector directions. Since PCA was calculated using all combined groups and time points, the eigenvector loadings remain the same between groups and across time resulting in similar modulation depth between groups. Therefore, differences in DC offsets are indicative of differences in PC scores between groups and across learning.

Minor concerns:

Line 48: should be “require” not “requires”

Fixed

Line 58: “enhanced signal-to-noise ratios within primary motor cortex” - this phase is vague and should be clarified

The text has been edited for clarity.

Line 77: “non-specific” - unclear what this means; suggest removing word

Removed

Line 172: “ Correlation coefficients from a covariance matrix” - nonstandard language, unclear what is meant here - the values in covariance matrices are not correlation coefficients.

Clarified in text, figure legends, and methods.

Line 178: It is not clear why the changes described just above (line 173: “widespread coactivation of corticospinal neuron pairs in the adaptive pull”) would not constitute refinement.

The corticospinal activation appeared to be non-specific during adaptive pull movements with a reduced number of corticospinal neurons selectively active during successful pulls in the late training stage, indicating even less refinement of the network over time.

Line 292: “These results confirm ...” - this claim is not supported; while it is true that the above results in the previous section on the inactivation experiments do support this claim, the fact that corticospinal neuron activation will in turn activate many other neurons in an aberrant way and disturb behavior does not specifically implicate corticospinal neurons per se in the task.

Indeed, while selective activation of C7/8 corticospinal neurons via channelrhodopsin will activate downstream motor centers, this does not diminish the role of patterned corticospinal activation in mediating precise movements. The selective corticospinal activation experiments show exactly that the timing of activity is critical for the evoked movements, regardless of the downstream substrates involved in movement execution.